

# New genera of Afrotropical Chalcidoidea (Hymenoptera: Cerocephalidae, Epichrysomallidae, Pirenidae and Pteromalidae)

Mircea-Dan Mitroiu[1], Jean-Yves Rasplus[2] and Simon van Noort[3,4]

[1] Faculty of Biology, Alexandru Ioan Cuza University of Iaşi, Iaşi, Romania
[2] CBGP, INRAE, CIRAD, IRD, Montpellier SupAgro, Université de Montpellier, Montpellier, France
[3] Research and Exhibitions Department, South African Museum, Iziko Museums of South Africa, Cape Town, South Africa
[4] Department of Biological Sciences, University of Cape Town, Cape Town, South Africa

## ABSTRACT

As a preliminary step towards the development of a key to genera of several families of Afrotropical Chalcidoidea, seven new genera in four families are described: Cerocephalidae–*Milokoa* Mitroiu, gen. nov. (type species: *Milokoa villemantae* Mitroiu, sp. nov.); Epichrysomallidae–*Delvareus* Rasplus, Mitroiu & van Noort, gen. nov. (type species: *Delvareus dicranostylae* Rasplus, Mitroiu & van Noort, sp. nov.); Pirenidae–*Afrothopus* Mitroiu, gen. nov. (type species: *Afrothopus georgei* Mitroiu, sp. nov.); Pteromalidae–*Kerangania* Mitroiu, gen. nov. (type species: *Kerangania nuda* Mitroiu, sp. nov.), *Pilosalis* Mitroiu, Rasplus & van Noort, gen. nov. (type species: *Pilosalis barbatulus* Mitroiu, sp. nov.), *Scrobesia* Mitroiu & Rasplus, gen. nov. (type species: *Scrobesia acutigaster* Mitroiu & Rasplus, sp. nov.), and *Spiniclava* Mitroiu & Rasplus, gen. nov. (type species: *Spiniclava baaiensis* Mitroiu & Rasplus, sp. nov.). Additionally, the following new species are described: *Pilosalis bouceki* Mitroiu & Rasplus, sp. nov., *Pilosalis eurys* Mitroiu & van Noort, sp. nov., *Pilosalis minutus* Mitroiu, sp. nov., *Pilosalis platyscapus* Mitroiu, Rasplus & van Noort, sp. nov., *Scrobesia pondo* Mitroiu, sp. nov., and *Spiniclava setosa* Mitroiu, sp. nov. All taxa are illustrated and the relationships with similar taxa are discussed. For each non-monotypic genus a key to species is provided.

## INTRODUCTION

The recent reclassification of Chalcidoidea based on a thorough phylogenomic hypothesis (*Burks et al., 2022*; *Cruaud et al., 2023*) has fundamentally transformed the circumscription of the family Pteromalidae *sensu Bouček (1988)*. In these works, twenty-three former subfamilies and tribes of Pteromalidae were elevated to family rank, and the family Pteromalidae now comprises only eight subfamilies and 415 genera in the world (*Burks et al., 2022*). During the preparation of the first key to the Afrotropical chalcidoid genera previously classified in Pteromalidae, a number of new taxa have been discovered; it is the

Corresponding author
Mircea-Dan Mitroiu,
mircea.mitroiu@uaic.ro

aim of this article to describe these new genera and species, in order to include them in the above-mentioned key.

The lack of keys to genera of most families of Afrotropical Chalcidoidea, and the fact that few comprehensive revisions of African genera have ever been published, are serious impediments for biodiversity studies. There is a great need for keys that will enable the investigation of the biology of the parasitoid species that could potentially be used as biocontrol agents against insect pests across Africa. A summary of the publications dealing with the Afrotropical Pteromalidae *sensu Bouček (1988)* was published by *Mitroiu (2011b)*. Since then, revisions of several genera have been published (*e.g.*, *Mitroiu, 2012*, *2015*, *2017*, *2019*, *2022*), but most genera remain uninvestigated. For the Afrotropical region, the Universal Chalcidoidea Database (*UCD Community, 2023*) lists 118 genera for the 19 subfamilies previously classified in Pteromalidae *sensu Bouček (1988)*. However, our on-going long-term study of African fauna has revealed a considerably higher number of taxa *i.e.*, over 200 genera, most of them described from other regions of the world, such as the Palaearctic or Australasian realms.

The difficulties in identifying the Afrotropical material of Pteromalidae partly arise from the lack of keys to Afrotropical and Neotropical genera, a rather limited understanding of the Australasian taxa despite the monumental work of *Bouček (1988)*, and a still incomplete revision of Risbec's and Masi's taxa (*Mitroiu, 2011a*). Many genera, previously known only from the Australasian region, also occur in the Afrotropical region (Mitroiu unpublished data). However, the taxonomic circumscription of these genera based on *Bouček (1988)* requires to be considerably extended to include the African taxa, which creates difficulty in making decisions about the correct placement of the African species. Thus, here we adopted a conservative approach and have delayed the description of several genera that we considered questionable regarding their taxonomic status.

## MATERIALS AND METHODS

The material described in this article is deposited in the following collections:

CBGP = Centre de Biologie pour la Gestion des Populations, Montpellier, France.

MNHN = Muséum national d'Histoire naturelle, Paris, France.

MICO = Mitroiu Collection, Alexandru Ioan Cuza University of Iași, Romania.

MRAC = Musée royal de l'Afrique centrale/Koninklijk Museum voor Midden-Afrika Tervuren, Belgium.

NHMUK = Natural History Museum, London, U.K.

NMPC = Natural History Museum, Prague, Czechia.

SAMC = South African Museum, Iziko Museums of South Africa, Cape Town, South Africa.

Classification follows *Burks et al. (2022)*. The morphological terminology follows *Gibson (1997)*. The antennal formula includes the 4th clavomere ("terminal button"), when visible. The body sculpture classification follows *Bouček & Rasplus (1991)*. Abbreviations of morphological terms are as follows:

fu = funicular segment.

gs = gastral sternite.

gt = gastral tergite.
H = height.
L = length.
LOL = lower ocular line.
MV = marginal vein.
OOL = ocellar-ocular line.
PMV = postmarginal vein.
POL = posterior ocellar line.
SV = stigmal vein.
W = width.

Images were either acquired at Alexandu Ioan Cuza University of Iași (CERNESIM laboratory, Iași, Romania) using a Leica DFC500 digital camera attached to a Leica M205A automated research stereomicroscope, or at INRAE using a Keyence digital microscope (VHX-500 Camera color CMOS and the VH-Z100UT lens). Focus stacking was performed with Zerene Stacker® and image clarity was enhanced using Adobe® Photoshop® 7.0.

Generic and species descriptions are generally concise and are focused on diagnostic characters. The holotype and opposite sex paratype (if available) are described and variation among other specimens is detailed separately, if necessary. All characters refer to females, if not stated otherwise. Information on specimen labels is given *ad litteram*.

Potentially new genera have been carefully assessed using the available generic keys (*Bouček, 1988*; *Bouček & Heydon, 1997*; *Bouček & Rasplus, 1991*; *Sureshan & Narendran, 2004*), as well as original descriptions for the genera not yet included in any key (mainly Neotropical taxa). The new genera were also compared with extensive material in the above-mentioned collections, as well as images from a comprehensive database containing photographs of Chalcidoidea. Potential relationships with similar taxa are extensively discussed for each genus. Within each family, the new genera and species are described in alphabetical order.

The electronic version of this article in Portable Document Format (PDF) will represent a published work according to the International Commission on Zoological Nomenclature (ICZN), and hence the new names contained in the electronic version are effectively published under that Code from the electronic edition alone. This published work and the nomenclatural acts it contains have been registered in ZooBank, the online registration system for the ICZN. The ZooBank LSIDs (Life Science Identifiers) can be resolved and the associated information viewed through any standard web browser by appending the LSID to the prefix http://zoobank.org/. The LSID for this publication is: urn:lsid:zoobank.org:pub:8A49E9CD-1FD9-4B3A-8285-CAA71CEE7A46. The online version of this work is archived and available from the following digital repositories: PeerJ, PubMed Central SCIE and CLOCKSS.

## RESULTS

Superfamily Chalcidoidea Latreille, 1817
Family Cerocephalidae Gahan, 1946

*Milokoa* Mitroiu, gen. nov.
urn:lsid:zoobank.org:act:8079D822-B567-4C69-844D-3329C6654618
(Fig. 1)

**Type species**
*Milokoa villemantae* Mitroiu, sp. nov., here designated.

**Diagnosis**

**Female**
Brachypterous (Fig. 1A); head with parascrobal area strongly inflated and with a pattern of two dark brown striate areas bordering a central patch of dense and white setation (Figs. 1B–1D); mesoscutum with strongly converging and incomplete notauli; axillae fused with mesoscutellum by broad striate band; propodeum strongly striate (Figs. 1F and 1H); lower mesepimeron raised above the surface of metapleuron and with conspicuous round convexity (Fig. 1G); gt6 strongly emarginate, exposing a large flat syntergum; cerci in dorsal position, with very long setae (Fig. 1A).

**Description**

**Female**
Body gracile, yellowish brown, without any metallic reflections, mainly smooth and glabrous except head (Fig. 1A).

Head triangular in frontal view and long in dorsal view (Figs. 1B and 1C). Clypeal margin almost straight (Fig. 1B). Tentorial pits present. Scrobal depression very deep, with strong interantennal crest continuing as a thin line until the upper margin of clypeus (Figs. 1B and 1C). Parascrobal area abruptly margined against scrobal depression and strongly inflated in lower part of the eye, with a pattern of two dark brown striate areas bordering a central patch of white dense setation (Fig. 1D). Gena not hollowed at mouth corner. Malar sulcus absent (Fig. 1D). Eyes moderately large, oval, glabrous, ventrally linearly diverging (Fig. 1B). Occiput with thin carina just before vertex continuing along posterior part of gena (Fig. 1C). Lower face and gena almost smooth, upper face mainly striate, scrobes much more finely so (Figs. 1B and 1C). Vertex mainly smooth (Fig. 1C). Head setation relatively dense but mostly inconspicuous except parascrobal areas where very conspicuous (Fig. 1B). Antennae inserted below LOL, toruli wide apart (Figs. 1B and 1C). Antenna moderately clavate, formula 11063 (Fig. 1E). Most funicular segments conical, with straight lateral margins (Fig. 1E). Clava pointed, segments closely fused (Fig. 1E). Scape fusiform (Fig. 1D). Mandibles small (number of teeth unknown).

Mesosoma elongated, moderately convex (Fig. 1G). Pronotum long conical, mostly striate, without any collar (Figs. 1F and 1G). Lateral side of pronotum flap-like, covering the base of fore coxa (Fig. 1G). Mesoscutum much wider than long, smooth (Fig. 1F). Notauli incomplete, strongly convergent and deep in anterior part and becoming more shallow and almost parallel posteriorly (Fig. 1F). Axillae not advanced, fused with mesoscutellum by a broad band of longitudinal striae (Figs. 1F and 1H). Mesoscutellum globose, triangular, smooth, with raised posterior border, frenum indicated (Figs. 1F and 1G). Metascutellum

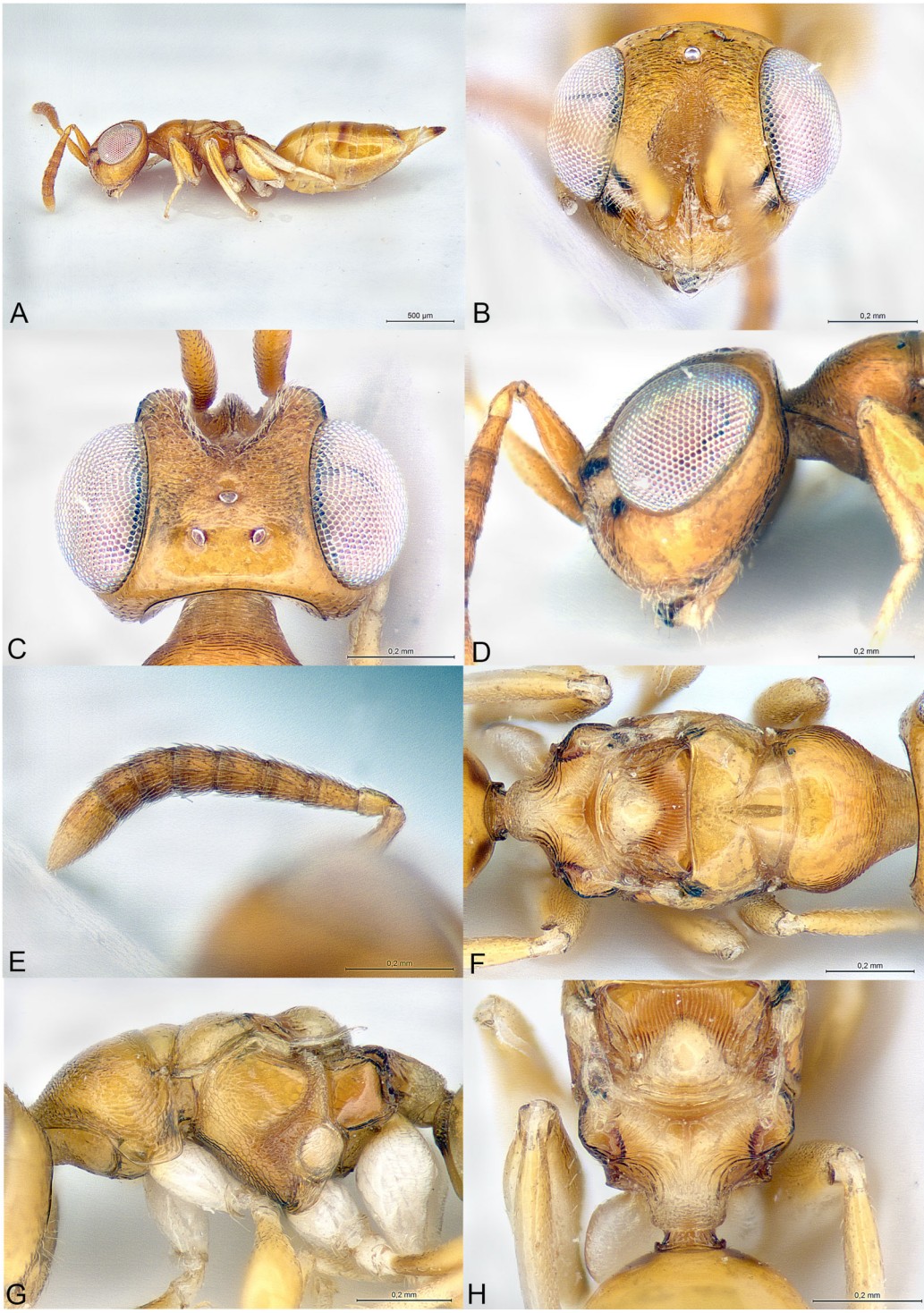

**Figure 1** *Milokoa villemantae.* (A) Female holotype, habitus, lateral. (B) Female holotype, head, frontal. (C) Female holotype, head, dorsal. (D) Female holotype, head, lateral. (E) Female holotype, antenna, lateral. (F) Female holotype, mesosoma, dorsal. (G) Female paratype, mesosoma, lateral. (H) Female holotype, propodeum, dorsal.

very short, vertical and smooth. Propodeum as a broad Y, with thin median carina and curved striae converging on large reticulate nucha (Fig. 1H). Posterior corners sharp (Fig. 1H). Plicae absent (Fig. 1H). Propodeal spiracle small, round, at considerable distance from anterior edge of propodeum, and adjacent to posterior smooth depression (Fig. 1H). Prepectus fairly large, mainly smooth, with posterior carina and almost reaching large tegula (Fig. 1G). Upper mesepimeron smooth, lower mesepimeron raised above the surface of metapleuron and with conspicuous shallowly reticulated round convexity (Fig. 1G). Mesepisternum with shallow reticulation (Fig. 1G). Metapleuron smooth (Fig. 1G). Hind coxa large, dorsally striate (Fig. 1G). Hind femur enlarged (Fig. 1A). Brachypterous (Fig. 1A). Fore wing narrow, reaching propodeal spiracle, venation barely visible (Figs. 1F and 1H). Hind wing indistinct.

Metasoma with petiole hardly visible, basally and ventrally embraced by short collar of gs1 (Fig. 1H). Gaster oval, dorsally convex (Fig. 1A), with gt1 moderately large, posterior margin broadly convex; gt6 medially very short, deeply emarginate to expose large setose and flat syntergum. Cerci on dorsal side of syntergum, near posterior margin of gt6, cercal setae very long (Fig. 1A). With a pair of spiracles on the lateral sides of gt6, adjacent to the posterior margin of gt5. Ovipositor sheaths moderately exserted (Fig. 1A).

**Male**
Unknown.

**Etymology**
From "miloko" meaning yellow in Malagasy (feminine gender).

**Relationships**
The family placement of this new genus was not as straightforward as expected and may even change in the future. Indeed, the specimens show similarities with Cerocephalidae, but also with Diparidae and Ceidae. *Bläser, Krogmann & Peters (2015)* suggested nine diagnostic characters for Cerocephalidae (as Cerocephalinae within Pteromalidae). Of these, six are characters bear by the wings, which are greatly reduced in the new genus and thus are not useful. The three remaining characters are: (1) interocular area with prominence (carina or tooth-shaped); present in the new genus as a sharp carina (Figs. 1B and 1C); (2) notauli complete; not true in the new genus as notauli are very superficial in the posterior part of mesoscutum and not reaching its hind margin (Fig. 1F); (3) hind tibia with two spurs; one spur is easily observable in the new genus, but the presence of the second is questionable as the distal extremity of the hind tibia bears several long and strong setae. According to *Burks et al. (2022)*, *Milokoa* generally fits our current family placement, based on the following features: (1) antenna with at most 10 flagellomeres, clava three-segmented (Fig. 1E); (2) intertorular prominence present (Figs. 1B and 1C); (3) mesoscutellum with frenum indicated (Fig. 1H); (4) acropleuron not expanded (Fig. 1G); (5) mesepimeron slightly extended over anterior margin of metapleuron (Fig. 1G). One character that appears different is the shape of the eyes, which are ventrally diverging in *Milokoa* and larger than in a typical cerocephalid (Figs. 1B and 1C).

There are only two genera of Cerocephalinae in which brachypterous females are encountered: *Chaetospilisca* Hedqvist, 1969 and *Theocolax* Westwood, 1832. Both share a characteristic head shape in frontal view, almost parallel sided, much higher than wide and with mandibular bases wide apart, which is not observed in *Milokoa*. Also, many other features of the antenna, mesosoma and metasoma in these two genera are very different from those found in *Milokoa*. In the key to world genera of Cerocephalinae (*Bläser, Krogmann & Peters, 2015*), assuming the fore wing disc is bare (a setose disc is found only in a fossil genus), the new genus runs to couplets 15 (if fore wing with a tuft of setae on parastigma) or 16 (if fore wing without a tuft of setae). As this character cannot be assessed because of wing reduction, the first case scenario leads to *Cerocephala* Westwood, 1832 (cosmopolitan), while the second leads to *Laesthiola Bouček, 1993* (Nearctic).

*Milokoa* shares the following characters with *Cerocephala*: head globose, with interantennal crest and raised parascrobal areas (Figs. 1B–1D); antenna 11063 (Fig. 1E); pronotum with flap-like lateral projections (Fig. 1G); propodeum with large nucha and smooth postspiracular foveae (Fig. 1H); gt6 emarginate. With *Laesthiola* it shares the following characters: head globose, with interantennal crest; antenna 11063; funicular segments with parallel sides (Fig. 1E); notauli strongly convergent (Fig. 1F); propodeum with nucha (*Bouček, 1993*).

*Milokoa* differs from both *Cerocephala* and *Laesthiola*, and from all other known cerocephalid genera by the following combination of features: (1) head with parascrobal area strongly inflated and with a pattern of two dark brown striate areas bordering a central patch of white dense setation (Figs. 1B and 1D); (2) mesoscutum with incomplete notauli (Fig. 1F); (3) axillae fused with mesoscutellum by broad striate band (Figs. 1F and 1H); (4) propodeum strongly striate (Fig. 1H); (5) lower mesepimeron with conspicuous round convexity (Fig. 1G); (6) gt6 strongly emarginate, exposing a large flat syntergum; (7) cerci in dorsal position, with very long setae (Fig. 1A).

The new genus also shows superficial similarities with some apterous Diparidae, such as a strongly modified mesosoma, striate hind coxae and long cercal setae. However, *Milokoa* differs from all known diparids in the antennal structure (Fig. 1E) (in Diparidae the antenna has 12 flagellomeres, including a 4th small clavomere), and from most diparids in the unexpanded gt1 (this state is found only in *Pyramidophoriella* Hedqvist, 1969 previously classified in Diparinae and currently *genus inquirendum* according to *Burks et al., 2022*), the raised mesepimeron (Fig. 1G) (found only in *Diparisca* Hedqvist), the lack of strong paired setae on dorsal side of head and mesosoma (Figs. 1C and 1F) (only six genera of diparids lack the strong paired setae, at least in some species), and the presence of a strong interantennal carina (Figs. 1B and 1C) (only a few genera without paired setae have a more or less strong interantennal carina). A comparison between the above diparid genera and *Milokoa* revealed several different character states based on the morphology of the head, mesosoma and metasoma.

The structure of the mesopleuron, having its hind margin conspicuously raised above the surface of the metapleuron (Fig. 1G), is reminiscent of the structure found in *Spalangiopelta* Masi, 1922 (Ceidae) and *Diparisca* Hedqvist, 1964 (Diparidae). However,

there are virtually no other characters that could suggest a relationship between these genera, except for the lower position of the toruli, superficial sculpture and brachypterism.

The head coloration pattern, with alternating brownish bands margining a white patch of setation (Figs. 1B and 1D), is similar to the pattern found in some species of *Eopelma* Gibson 1989 (Eupelmidae), such as *E. gibsoni* Fusu and Polaszek, 2017, or *Dipara* Walker, 1833 (Diparidae), such as *D. nyani* *Braun & Peters (2021)*. To a lesser degree it is also similar to the pattern found in some *Pseudoceraphron* Dodd, 1924 (Neapterolelapinae, *incertae sedis*), such as *P. belissimus* Jałoszyński, 2020. These similarities may indicate a convergence due to an unknown ecological function in parasitoids dwelling in leaf litter.

Many of the unique characters of *Milokoa* are probably related to apterism (mesosoma structure) and adult emergence and/or host location (head structure), as observed in other Chalcidoidea.

### *Milokoa villemantae* Mitroiu, sp. nov.
urn:lsid:zoobank.org:act:B78D19D2-C6FE-49F5-811F-FC6126E964D6
(Fig. 1)

**Material examined**

**Holotype**

**MADAGASCAR:** ♀, "Madagascar: Namoroka, 25-27/10/2016, YPT no 5B, C. Villemant rec"; EY36195 (MNHN).

**Paratype**

**MADAGASCAR:** 1♀, "Madagascar: Namoroka, 23-25/10/2016, YPT no 5B, C. Villemant rec.", MICO-2023-1 (MICO).

**Description**

**Female holotype**
Body length: 2.3 mm. Colour as in Fig. 1. Interantennal crest strong, blade-like (Figs. 1B and 1C), but not protruding over the inflated parascrobal areas, not visible in lateral view of the head (Fig. 1D). Apart from the large patches of white setae (Figs. 1B and 1D), parascrobal area densely setose along lateral margins of scrobal depression, setae becoming sparser towards the eye and vertex. Ocelli in an almost equilateral triangle (Fig. 1C). Antennal sensilla in one sparse row on each funicular segment, difficult to observe among dense setation (Fig. 1E). Fore wing reduced and just covering propodeal spiracle (Figs. 1F and 1H). Propodeum extensively striate and with small smooth central area (Fig. 1H). Propodeal spiracle separated from anterior margin of propodeum by about 3X its diameter. Postspiracular smooth depression oval and reaching posterior margin of propodeum (Fig. 1H). Relative measurements: Head L: 37, W: 59, H: 52; POL: 8; OOL: 8; eye H: 33, L: 25; eye L dorsally: 26; temple L dorsally: 6; malar space: 17; mouth W: 22; scape L: 28, W 6; pedicel L: 8, W: 4; pedicel plus flagellum L: 65; fu1 L: 10, W: 5.5; fu6 L: 7, W: 8; clava L: 15, W: 8. Mesosoma L: 85, W: 40, H: 38; mesoscutum L: 20, W: 40;

mesoscutellum L: 20, W: 19; propodeum L: 20; fore wing L: 20, W: 5. Metasoma L: 118, W: 51; gt1 L: 30, W: 46; gt6 L: 2, W: 30; syntergum L: 15, W: 22.

**Variation**
Body length: 2.1–2.3 mm.

**Etymology**
The species is dedicated to Claire Villemant (MNHN), who collected the type material of the new species (noun in genitive case).

**Distribution**
Madagascar.

**Biology**
Unknown.

Family Epichrysomallidae Hill and Riek, 1967

***Delvareus* Rasplus, Mitroiu & van Noort, gen. nov.**
urn:lsid:zoobank.org:act:D4085947-BE16-44F9-9502-692B31FDA24F
 (Fig. 2)

**Type species**
*Delvareus dicranostylae* Rasplus, Mitroiu & van Noort, **sp. nov.**, here designated.

**Diagnosis**

**Female**
*Delvareus dicranostylae* is immediately recognizable by the pectinate antenna, bearing seven rami (six on funiculars and one on the first clavomere) (Fig. 2B); last clavomeres fused subtriangular and widening distally, bilobed at the extremity. Wings hyaline and subglabrous, with sparse inconspicuous dot-like setae. MV more than 1.5X SV. Notauli only indicated by darker internal ridge and reaching the transscutal articulation inside of scutoscutellar sutures.

**Description**

**Female**
Body robust, black and yellowish on antero-lateral part of pronotum, legs yellow except proximal half of metacoxa blackish (Fig. 2A). Body setation very short and scattered except on mesosternum and a few setae on propodeal callus.

   Head in frontal view strongly transverse, about 2.2X as wide as long (Fig. 2D). Clypeal margin very slightly bilobed (Fig. 2D). Tentorial pits present. Scrobal depression shallow, inconspicuous. Malar sulcus shallow, hardly traceable near the eye. Occiput with conspicuous occipital carina (Fig. 2E). Head smooth, except clypeus and lower face, which are mostly alutaceous. Antennal insertion well above LOL, just below the middle of face (Fig. 2D). Antennal formula 11061 (Fig. 2B). No anellus visible. Antennal scape normal. The six funicular segments bearing a curved and long ramus, transversely striped.

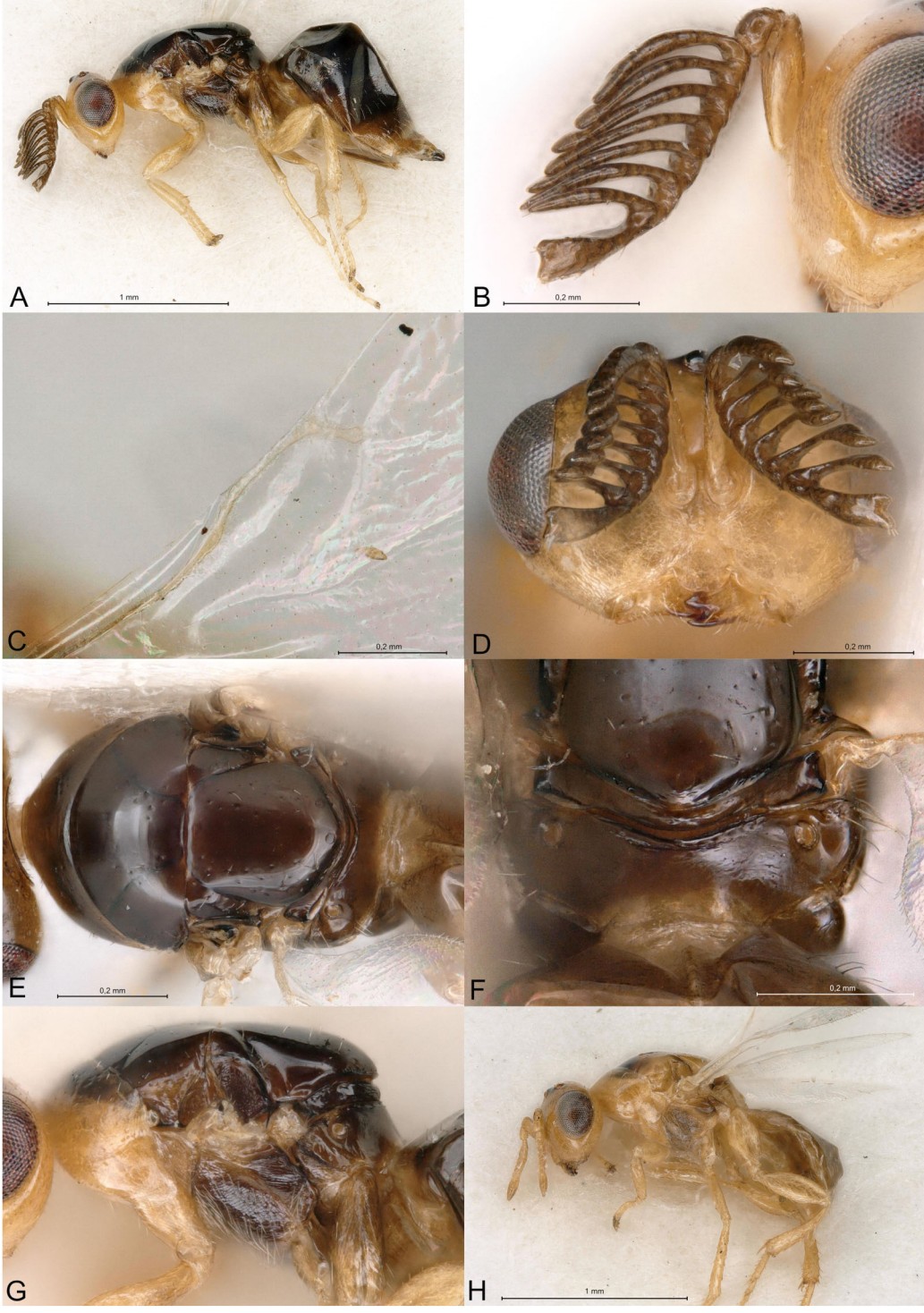

**Figure 2** *Delvareus dicranostylae.* (A) Female paratype, habitus, lateral. (B) Female paratype, antenna, lateral. (C) Female paratype, fore wing, dorsal. (D) Female paratype, head, frontal. (E) Female paratype, mesosoma, dorsal. (F) Female paratype, propodeum, dorsal. (G) Female paratype, mesosoma, lateral. (H) Male paratype, habitus, lateral.

Antennal clava bearing a basal ramus, terminal part subtriangular, elongated, widening distally and bilobed, each lobe bearing sensilla. Mandibles not enlarged, with three teeth.

Mesosoma convex (Fig. 2A). Pronotum short, without pronotal collar (Fig. 2E). Mesonotum smooth, with a few piliferous punctures posteriorly (Fig. 2E). Notauli obsolete, only traceable anteriorly (Fig. 2E). Mesoscutellum convex, smooth with 2–4 setae on side; frenal line absent (Fig. 2E). Mesoscutellum broadly bordering mesoscutum; scutoscutellar suture abutting transscutal articulation externally to dark internal ridges of notauli (Fig. 2E). Metascutellum extremely short and smooth. Propodeum short, flattened, entirely smooth, without median carina (Fig. 2F). Propodeal spiracles large with a conspicuous rim internally, but flap-like expansion, external of propodeal spiracle, absent (Fig. 2F). Prepectus reticulated, large, longer than tegula. Mesepisternum, mesopleuron and metapleuron finely reticulate. Mesepisternum and lower mesepimeron with white, long and dense setae (Fig. 2G). Hind tibia with one spur. All legs with five tarsomeres. Fore wing (Fig. 2C) hyaline, setation extremely sparse and short, dot-like; fringe absent. Marginal vein not widened, 1.8 times the length of stigmal vein, which is 4X as long as postmarginal vein. Stigmal oblique, forming a 60° angle with postmarginal. Stigma moderately capitate.

Metasoma with petiole short, virtually inconspicuous. Gaster high, dorsally curved, slightly shorter than head plus mesosoma (Fig. 2A). Posterior margin of gt1 slightly emarginated medially. Syntergum narrower than previous tergite and pointed. Hypopygium large, extending beyond ¾ of gaster length (Fig. 2A). Cercus elongate with all setae equal. Ovipositor sheaths short (Fig. 2A).

**Male**
Similar to female, but antenna filiform, without any rami (Fig. 2H). Antennal formula 11151. First funicular segment shorter and narrower than following ones, subtriangular. Gaster shorter, not dorsally curved (Fig. 2H).

**Etymology**
The genus (masculine gender) is dedicated to our colleague and friend Gérard Delvare (CIRAD), who kindly gave us the specimens of this new genus.

**Relationships**
Among Epichrysomallidae genera, the new genus is closely related to *Acophila* Ishii, 1934, which occurs mostly in the Oriental and Australian regions, with only few undescribed species in the Afrotropics. Both genera are characterized by the presence of an external occipital carina; notauli inconspicuous, only visible anteriorly and not reaching the transscutal articulation; mesoscutellum widely abutting the transscutal articulation; presence of only five or six funicular segments. *Delvareus* is easily separated from *Acophila* by its pectinate antennae bearing seven long rami (Fig. 2B) (filiform in *Acophila*); its transverse head (Fig. 2D), 1.6X as wide as high *versus* at most 1.1–1.2X as wide as high in Afrotropical species of *Acophila*; the female formula antenna 11061 with no anellus (Fig. 2B) *versus* 11153 in *Acophila*.

Finally, *Sycotetra* Bouček, 1981 contains one undescribed species with pectinate antennae in Africa, which could be confused with *Delvareus*. However, this *Sycotetra* species, associated with *Ficus natalensis*, can be easily separated from *Delvareus* by the following characters: first two funiculars without any rami (the antennae exhibit only four rami that are further covered with long sensilla); all tarsi tetramerous; and gaster dorsally keeled and strongly compressed laterally.

### *Delvareus dicranostylae* Rasplus, Mitroiu & van Noort sp. nov.

urn:lsid:zoobank.org:act:5E6F6D56-470C-4FA5-B3FA-C95DB47D8664

(Fig. 2)

**Material examined**

**Holotype**

**BÉNIN:** ♀, "Bénin Rte N'Dali-Ina, 10.xi.1993 Delvare G., ex *Ficus* sp.", JRAS01442_0101 (CBGP).

**Allotype**

**BÉNIN:** ♂, as holotype, JRAS01442_0102 (CBGP).

**Additional paratypes**

**BÉNIN:** 7♀, JRAS01442_0103 to JRAS01442_0109 (CBGP); 2♂, as holotype, JRAS01442_0110, JRAS01442_0111 (CBGP).

**Description**

**Female holotype**

Body length: 2.5 mm. Colour as in Fig. 2A. Head transverse, 1.6X wider than high. Clypeus 1.9X as broad as high. Clypeal margin slightly bilobed (Fig. 2D). Supraclypeal area small, subrectangular, 0.8X as wide as diameter of median torulus, slightly delimited by shallow groove. Scrobes shallow. Antenna inserted well above LOL, near the center of face (Figs. 2B and 2D). Scape subcylindrical, 3X as long as wide and 3.3X as long as pedicel, not reaching ventral margin of median ocellus. Pedicel as long as wide. Clava 5X as long as wide and 4.4X as long as last funicular segment. Malar sulcus present but faint. Mesosoma dorsally smooth, with scattered piliferous punctures (Fig. 2E). Pronotum 0.29X as long as mesonotum. Mesoscutum with a few short setae. Mesoscutellum 0.92X as wide as long and 1.2X as long as mesoscutum, with few scattered short setae. Propodeum entirely smooth (Fig. 2F). Fore wing subglabrous with only sparse dot-like setae, fringe absent (Fig. 2C). Relative measurements. Head L: 48, W: 104, H: 65; eye H: 40, L: 23; malar space: 24; mouth W: 48; scape L: 30, W: 9; pedicel L: 9, W: 9; pedicel plus flagellum L: 85. Mesosoma L: 133, W: 93, H: 85; pronotum L: 15, W: 88; mesoscutum L: 51, W: 93; mesoscutellum L: 61; W: 56; propodeum L: 19, W: 82; fore wing L: 276, W: 120; MV: 31; SV: 17; PMV: 4. Metasoma L: 161, W: 98; gt1 L: 42, W: 98; gt6 L: 10, W: 76; syntergum L: 5, W: 14.

**Male allotype**
Length 1.5 mm. Colour as in Fig. 2H. Head 1.6X wider than high. Flagellomeres without rami (Fig. 2H), transverse except F1 subtriangular, 1.1X as long as wide and 0.36X as long as pedicel. Clava undivided, 2.1X as long as wide and 5.8X as long as last funicular segment. Gena 0.5 x length of eye. Malar sulcus absent.

**Variation**

**Female**
Body length: 2.1–2.5 mm.

**Etymology**
The name of the species (noun in genitive case) refers to the probable host fig of this species, *Ficus dicranostyla* Mildbr. (Moraceae).

**Distribution**
Bénin.

**Biology**
Specimens were obtained from figs of an unidentified fig tree together with specimens of *Dolichoris flabellatus* Wiebes, 1979 (Hymenoptera: Agaonidae). This pollinating wasp is known to be associated with *Ficus dicranostyla* and *F. variifolia* Warb. in tropical Africa. These two species belonging to subgenus *Oreosycea* are suspected to just be conspecific ecotypes. The small-leaved tree (*F. dicranostyla*) occurs in savanna woodlands on rocks while the tree with large and variable shaped leaves (*F. variifolia*) occurs in lowland and evergreen forests. The dry habitats of northern Bénin, where the new epichrysomalid genus has been sampled, host only *F. dicranostyla*, which strongly suggests that this species is its host fig.

Family Pirenidae Haliday, 1844
Subfamily Tridyminae Thomson, 1876

***Afrothopus* Mitroiu, gen. nov.**
urn:lsid:zoobank.org:act:6DCDAD4A-03E5-4B98-814B-355B8A93B97E
 (Figs. 3, 4)

**Type species**
*Afrothopus georgei* Mitroiu, sp. nov., here designated.

**Diagnosis**

**Both sexes**
Head and mesosoma coarsely reticulated, with bright metallic reflections (Figs. 3A–3G); antenna inserted slightly below LOL (Fig. 3B); clypeal margin convex (Figs. 3B–3D); short interantennal crest present (Figs. 3B–3D); pronotum with large diverging shoulders, collar medially steep and short, without carina (Figs. 3E and 3F); mesoscutum long (Fig. 3E); notauli complete, thin and shallow (Fig. 3E); propodeum with median carina and nuchal

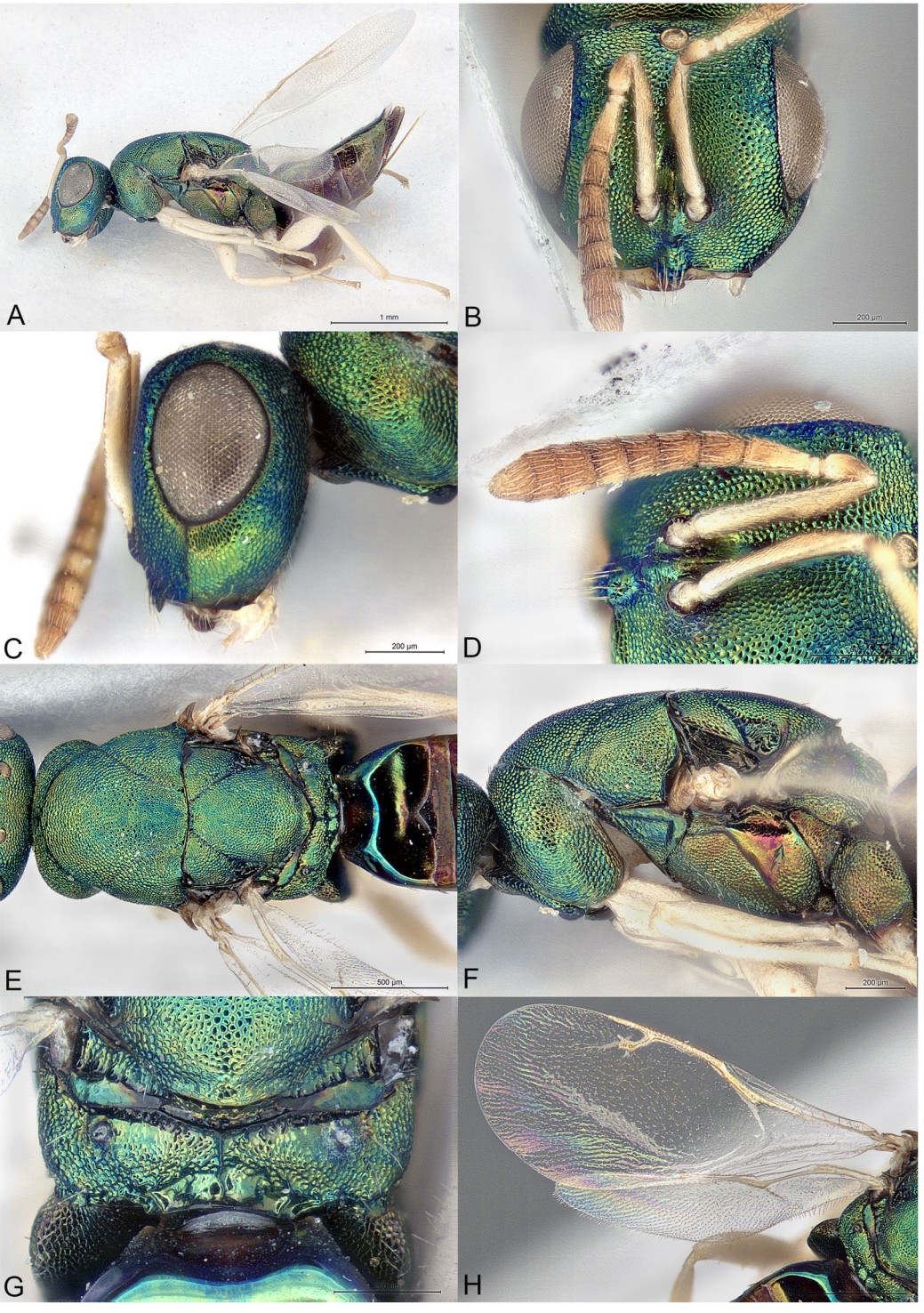

**Figure 3 *Afrothopus georgei.*** (A) Female holotype, habitus, lateral. (B) Female holotype, head, frontal. (C) Female holotype, head, lateral. (D) Female holotype, antenna, lateral. (E) Female holotype, mesosoma, dorsal. (F) Female holotype, mesosoma, lateral. (G) Female holotype, propodeum, dorsal. (H) Female holotype, fore and hind wings, dorsal.

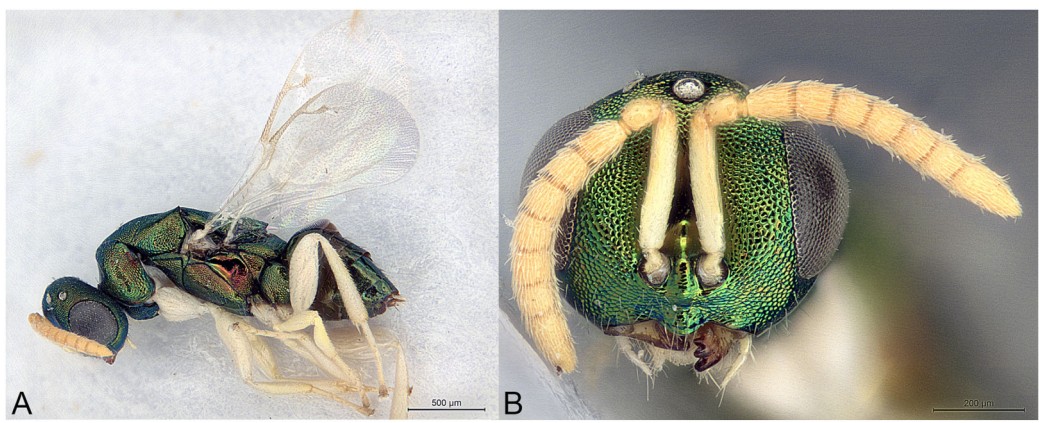

**Figure 4 _Afrothopus georgei._** (A) Allotype male, habitus, lateral. (B) Allotype male, head, frontal.

strip (Fig. 3G); fore wing without fringe (Fig. 3H); gt1 with posterior margin strongly incised in the middle (Fig. 3E).

**Female**
Upper face conspicuously raised near internal upper eye margin, with regular reticulation (Figs. 3B and 3C); clypeal margin with central lobe bearing several setae (Figs. 3B and 3D); antenna clavate, 11-segmented, with one microscopic anellus (not counted), one large anellus and five funicular segments, antennal formula 11154 (Fig. 3D); hypopygium in anterior half of gaster (Fig. 3A).

**Male**
Upper face normal (Fig. 4B); clypeal margin broadly convex (Fig. 4B); antenna filiform, 12-segmented, with one visible anellus and six broad funicular segments, formula 11163 (Fig. 4B).

**Description**

**Female**
Body metallic (Figs. 3A–3G). Setation mostly absent or inconspicuous, except on lower head and propodeal callus.

Head in frontal view approximately round (Fig. 3B). Upper face conspicuously raised near internal upper eye margin (Figs. 3B and 3C). Clypeal margin with central lobe covered by long setae (Fig. 3B). Tentorial pits absent (Fig. 3B). Scrobal depression shallow, with short interantennal crest (Figs. 3B–3D). Gena not hollowed at mouth corner (Fig. 3C). Malar sulcus present (Fig. 3C). Eyes slightly linearly diverging in lower part (Fig. 3B). Temples strongly converging in dorsal view of the head. Occiput without carina. Antennal insertion slightly below LOL, 11154 (Fig. 3D). First anellus microscopic, the second much larger. Antennal clava symmetric, without conspicuous area of microsetation, distal end rounded (Fig. 3D). Mandibles not unusually large.

Mesosoma convex (Fig. 3F). Pronotum with large diverging shoulders (Fig. 3E). Collar medially steep and short, without carina (Fig. 3F). Mesoscutum long (Fig. 3E). Notauli complete, thin and shallow (Fig. 3E). Axillae very slightly advanced (Fig. 3E). Mesoscutellum convex, posterior margin slightly expanded (Fig. 3G). Frenal line absent, but frenum slightly indicated (Fig. 3G). Propodeum (Fig. 3G) short. Plicae absent, indicated only on lateral sides of smooth nuchal strip. Median carina present. Propodeal hind corners not prominent and not sharp. Propodeal spiracles small, almost touching posterior margin of metanotum. Prepectus very large, uniformly sculptured (Fig. 3F). Fore and hind legs strong (Fig. 3A). Hind coxa large triangular, dorsally bare. Hind tibia with two unequal spurs. Fore wing (Fig. 3H) hyaline. Fore wing basally bare, fringe absent. Marginal vein not widened. Stigmal vein much shorter than marginal vein, stigma moderately capitate. Postmarginal vein much shorter than marginal vein and slightly longer than stigmal vein.

Metasoma with petiole inconspicuous. Gaster oval, dorsally flat (Fig. 3A). Gt1 the largest, its posterior margin broadly incised and hence appearing bilobed (Fig. 3E). Syntergum small, broader than long. Hypopygium large, in the anterior third of gaster. Cercal setae equal. Ovipositor sheaths short (Fig. 3A).

### Male
Similar to female (Fig. 4A), except mainly for the differential features given in the diagnosis.

### Etymology
The genus name (masculine) is derived from Africa and the suffix -*thopus*, indicating some affinities with *Spathopus* Ashmead.

### Relationships
The family placement of *Afrothopus* first appeared difficult, the general habitus indicating placement in the family Pteromalidae *sensu lato*. However, a careful examination strongly suggested that this new genus was best placed in the family Pirenidae, subfamily Tridyminae, based on the following characters: (1) antenna with only 10 visible flagellomeres in female and 11 in male, with five (female) or six (male) large flagellomeres and one anelliform flagellomere (plus a microscopic one) (Figs. 3D and 4B); (2) eyes slightly linearly diverging (Figs. 3B and 4B); (3) clypeus without transverse apical groove, with median convexity (Figs. 3B, 3D and 4B); (4) notauli complete (Fig. 3E); (5) marginal vein less than 3X stigmal vein (Figs. 3H and 4A).

The female *Afrothopus* has the following unique combination of characters among Pirenidae: (1) upper face conspicuously raised near internal upper eye margin, with regular reticulation (Figs. 3B and 3C); (2) clypeal margin with small central lobe bearing several setae (Figs. 3B and 3D); (3) short interantennal crest (Figs. 3B–3D); (4) head and dorsal side of mesosoma reticulated (Figs. 3B and 3E); (5) pronotum with large diverging shoulders, collar medially steep and short, without carina (Figs. 3E and 3F); (6) mesoscutum long, notauli thin and shallow (Fig. 3E); (7) fore wing without fringe (Fig. 3H); (8) gt1 with posterior margin strongly incised in the middle (Fig. 3E).

In the generic key to Palaearctic Pteromalidae (*Bouček & Rasplus, 1991*) *Afrothopus* runs to couplet 290 (*Melancistrus* Graham 1969 and *Gastrancistrus* Westwood 1833). The new genus appears closer to *Gastrancistrus*, as the hypopygium does not end in a narrow projection and the propodeum lacks a transverse crest. However, *Afrothopus* differs from *Gastranscistrus* in most of its diagnostic characters, except for the general features of Pirenidae (see above).

In the generic key to Nearctic Pteromalidae (*Bouček & Heydon, 1997*) the new genus runs to couplet 38 and *Spathopus* Ashmead, 1904 based on the diverging anterior corners of pronotum. Other similarities with *Spathopus* are the shape of the lower face, including the presence of a small interantennal crest and the shape of the clypeal margin, with a central convex lobe. Beside other characters being different in the new genus (see diagnosis), the antenna differs from that of *Spathopus* in having a conspicuous anellus (in *Spathopus* the anellus is inconspicuous, the antenna having only 10 visible segments). This situation is also encountered in *Ecrizotes* Förster, 1861, where there are no visible anelli. At the same time the females of *Ecrizotes* have five large segments between pedicel and clava, while the males have six, as in *Afrothopus*. The new genus differs from *Ecrizotes* in most of its diagnostic features, except for the characters that are shared with other Pirenidae, and the similar antenna. Moreover, all known species of *Spathopus* and *Ecrizotes* are black or have at most dark metallic reflections, and the hypopygium is situated at or even beyond the posterior extremity of gaster.

In the generic key to Australasian Pteromalidae (*Bouček, 1988*) females of *Afrothopus* run to couplet 300 (*Amuscidea* Girault, 1913 and *Gastrancistrus*). These two genera are closely related, the only difference being their mandible formula: 3:3 in *Amuscidea* and 4:4 (rarely 3:4) in *Gastrancistrus* (*Bouček, 1988*). Unfortunately, in all specimens of the type series of *Afrothopus* the mandibles are held in a closed position, except for the left mandible of a male, which has three teeth. The differences between *Gastrancistrus* and *Afrothopus* are discussed above.

In the generic key to Oriental Pteromalidae (*Sureshan & Narendran, 2004*) the new genus runs to couplet 32 (*Gastrancistrus* and *Trigonoderopsis* Girault 1915). *Trigonoderopsis* greatly differs from *Afrothopus* in many features (female antenna with six funicular segments, a much longer marginal vein, different head shape, different mesosoma, *etc.*), and is now classified in Colotrechninae: Trigonoderopsini (Pteromalidae) (*Burks et al., 2022*).

Most Pirenidae have the head and the dorsal side of mesosoma smooth or weakly reticulated. The exceptions are *Watshamia* Bouček, 1974 (Afrotropical) and *Velepirene Bouček (1988)* (Australasian). Both these genera are close to *Macroglenes* Westwood, 1832, are classified in the subfamily Pireninae, and thus are very different from *Afrothopus*.

The head of the female *Afrothopus* has some similarities with the head of *Tanina* Bouček, 1976 (Pteromalinae), *i.e.*, the face is distinctly swollen near the inner eye margin. We hypothesize that this feature is related to adult emergence, oviposition or host searching activity; together with the moderately deep scrobes and the presence of the

interantennal crest, this character suggests a mechanism for the protection of the antennae during such activities.

### *Afrothopus georgei* Mitroiu, sp. nov.
urn:lsid:zoobank.org:act:CC8CB6B6-F273-4BC9-B0D5-0932DDA1114F
(Figs. 3, 4)

**Material examined**

**Holotype**

ZIMBABWE: ♀, "Rhodesia: Chishawasha, ix. 1979, A. Watsham", NHMUK014444237 (NHMUK).

**Allotype**

ZIMBABWE: ♂, "Zimbabwe: Chishawasha, vii. 1979, A. Watsham", NHMUK014444238 (NHMUK).

**Additional paratypes**

ZIMBABWE: 1♂, "Zimbabwe: Salisbury, Jan. 81, A. Watsham", NHMUK014444239 (NHMUK); 1♂ "Zimbabwe: Chishawasha, nr. Salisbury, viii. 1978, A. Watsham", MICO-2023-2 (MICO).

**Description**

**Female holotype**
Body length: 3.00 mm. Colour as in Fig. 3. Central lobe of the clypeal margin narrow, with several conspicuous setae (Figs. 3B and 3D). Head, including projection adjacent to inner eye margin, mostly uniformly and coarsely reticulate (Figs. 3B–3D). Antenna (Fig. 3D) distinctly clavate. Second anellus much larger than the first, which is microscopic. First funicular segment long conical, basally narrower than pedicel. Sensilla thin, in one row on all funiculars. Most of the dorsal side of mesosoma uniformly and coarsely reticulate (Fig. 3E). Mesoscutellum with frenal area indicated by a very slight change in sculpture (Fig. 3G). Posterior part of axilla and axillula more irregularly sculptured. Propodeum (Fig. 3G) uniformly reticulate except straight median carina reaching posterior margin of propodeum and shiny nuchal strip. Prepectus, mesepisternum and metapleuron uniformly reticulate (Fig. 3F). Upper mesepimeron almost smooth, separated from reticulate lower mesepimeron by an incomplete groove (Fig. 3F). Fore wing (Fig. 3H) extensively bare in basal half. Basal cell bare. Speculum reaching stigmal vein. Area between stigmal and postmarginal veins bare. Relative measurements: Head L: 40, W: 73, H: 62; eye H: 39, L: 28; malar space: 20; mouth W: 39; scape L: 37, W 6; pedicel L: 8, W: 6.5; pedicel plus flagellum L: 70; fu1 L: 12, W: 6; fu5 L: 8, W: 9; clava L: 19, W: 10. Mesosoma L: 120, W: 71, H: 65; mesoscutum L: 59, W: 71; mesoscutellum L: 50, W: 44; propodeum L: 15; fore wing L: 187, W: 80; MV: 37; SV: 15; PMV: 21. Metasoma L: 120, W: 60; gt1 L: 35, W 58; gt6 L: 10, W: 35; syntergum L: 5, W: 15.

**Male allotype**

As the female, except mainly the following. Colour as in Fig. 4. Head without any projection adjacent to inner eye margin (Fig. 4B). Convexity of the clypeal margin less narrow, arch-like (Fig. 4B). Eye rounder. Antenna (Fig. 4B) less clavate and more densely setose, with six funiculars. Both anelli extremely small. Proximal funiculars wider, the first conspicuously wider than pedicel, length about 1.2X width. Mesoscutum with several piliferous punctures among reticulation. Gaster much shorter than mesosoma (Fig. 4A), length about 1.5X width.

**Variation**

**Males**

Body length: 2.00–2.25 mm. Head and mesosoma with the coppery reflections more or less obvious. Antennae and legs from whitish to yellow. Pedicel sometimes infuscate basally. Gaster length 1.5–2.0X width, depending on the degree of collapse.

**Etymology**

The new species is named after George, the son of Mircea-Dan and Simona (noun in genitive case).

**Distribution**

Zimbabwe.

**Biology**

Unknown.

Family Pteromalidae Dalman, 1820
Subfamily Pteromalinae Dalman, 1820
Tribe Pteromalini Dalman, 1820

**_Kerangania_ Mitroiu, gen. nov.**

urn:lsid:zoobank.org:act:DC6BE0D6-F230-4E97-A057-F814DC691140
   (Fig. 5)

**Type species**

_Kerangania nuda_ Mitroiu, sp. nov., here designated.

**Diagnosis**

**Female**

Body black, without metallic reflections (Figs. 5A–5H); head anteroposteriorly short (Fig. 5A); clypeal margin bilobed (Fig. 5C); occipital carina present (Fig. 5E); maxillary palpus unusually long and thin (Figs. 5B, 5C and 5F); pronotum separated from lateral lobes of mesoscutum by deep groove; metascutellum extremely short, as a smooth line (Fig. 5G); propodeum short, median area convex, reticulate, with indication of oblique costula (Fig. 5G); petiole extremely short and wide (Fig. 5G); fore wing setation pale, inconspicuous, fringe absent (Fig. 5H); ovipositor sheaths long (Fig. 5A).

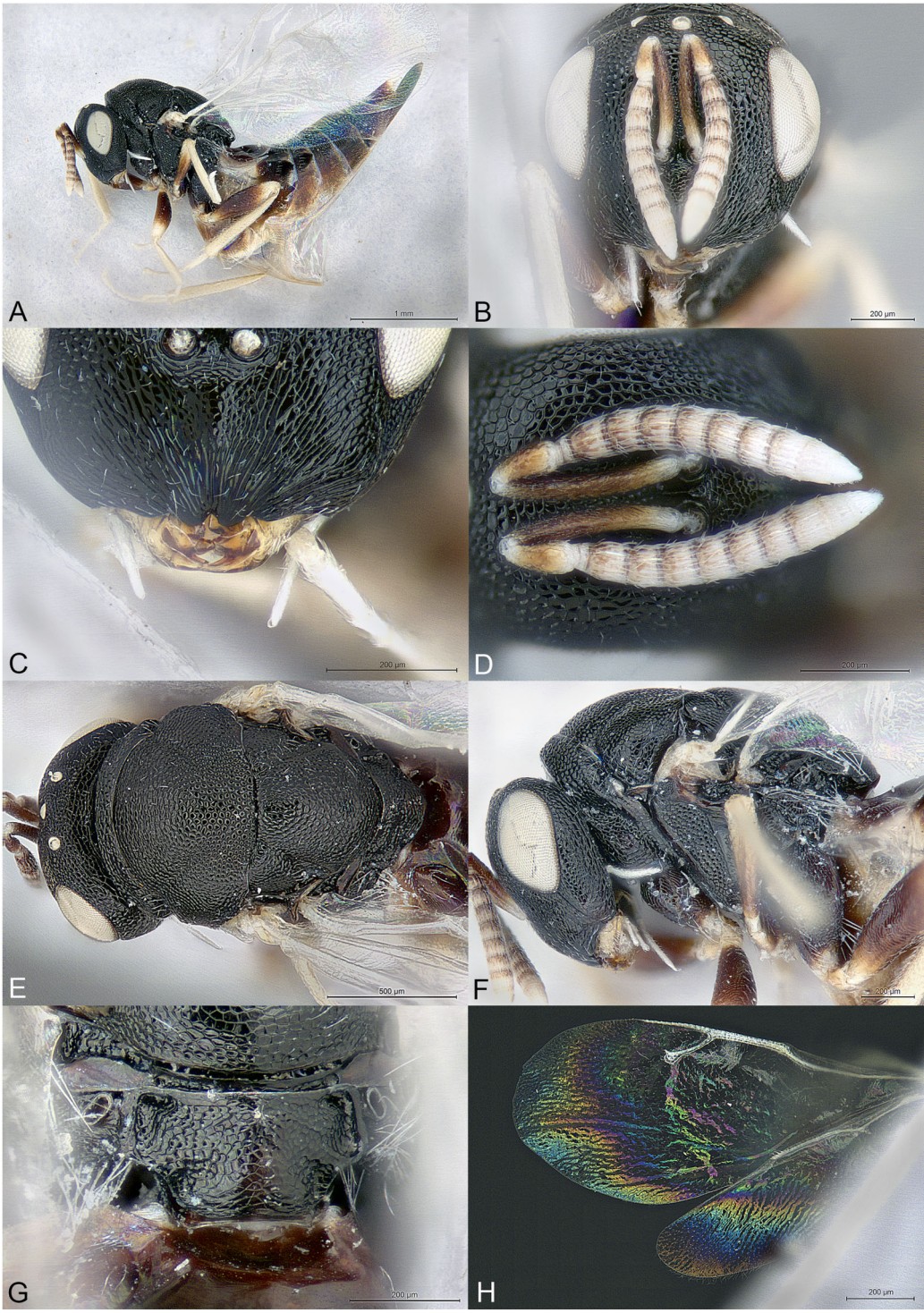

**Figure 5** *Kerangania nuda.* (A) Female holotype, habitus, lateral. (B) Female holotype, head, frontal. (C) Female paratype, clypeus. (D) Female holotype, antennae, lateral. (E) Female holotype, mesosoma, dorsal. (F) Female holotype, mesosoma, lateral. (G) Female holotype, propodeum, dorsal. (H) Female holotype, fore and hind wings, dorsal.
## Description

### Female
Body robust, black, without any metallic reflections (Figs. 5A–5G). Body setation mostly absent or inconspicuous except on propodeal callus.

Head in frontal view almost round (Fig. 5B), anteroposteriorly short (Fig. 5A). Clypeal margin bilobed (Fig. 5C). Tentorial pits absent. Scrobal depression moderately deep, clearly visible in dorsal view of the head. Gena not hollowed at mouth corner (Fig. 5F). Malar sulcus shallow (Fig. 5F). Eyes normal. Occiput with strong carina, conspicuous in dorsal view of the head (Fig. 5E). Head sculpture (Figs. 5B–5D) mostly reticulate, except clypeus and lower face, which are mostly striate. Antennal insertion above LOL, approximately in the middle of face (Figs. 5B and 5C). Antennal formula 11264 (Fig. 5D). Both anelli transverse. Antennal scape normal. Antennal clava symmetric, with small area of microsetation, distal end rather acute but not pointed. Mandibles not large. Maxillary palpus with terminal segment long, thin and setose (Figs. 5B, 5C and 5F). Labial palpus normal.

Mesosoma convex (Fig. 5F). Pronotum short, separated from lateral lobes of mesoscutum by deep groove (Figs. 5E and 5F). Pronotal collar present, anterior margin abrupt but not carinate (Fig. 5F). Notauli incomplete, very superficial, extending on more than half the length of mesoscutum (Fig. 5E). Axillae slightly advanced. Mesoscutellum convex, frenal line absent (Fig. 5E). Sculpture of mesoscutum and mesoscutellum mostly uniformly reticulate (Fig. 5E). Metascutellum extremely short, as a smooth line (Fig. 5G). Propodeum (Fig. 5G) short, convex, uniformly reticulate. Plicae well defined and reaching the short nucha. Median carina absent. Costula slightly indicated, oblique. Propodeal hind corners prominent posteriorly but not sharp and without any carinae. Propodeal spiracles large, oval, virtually touching the metanotum. Prepectus smooth, much shorter than tegula (Fig. 5F). Mesopleuron with upper mesepimeron largely smooth and mesepisternum uniformly reticulate (Fig. 5F). Metapleuron finely reticulate. Legs slender (Fig. 5A). Hind coxa dorsally bare except several long setae. Hind tibia with one spur. Fore wing (Fig. 5H) hyaline. Wings setation extremely thin and pale, visible only against a dark background. Fore wing basal third mostly bare. Fringe present only on hind wing. Marginal vein not widened. Stigmal vein shorter than both marginal and postmarginal veins. Stigma moderately capitate. Postmarginal vein shorter than marginal vein.

Metasoma with petiole extremely short and wide, not or hardly visible under nucha (Fig. 5G). Gaster lanceolate, dorsally flat, longer than head plus mesosoma (Fig. 5A). Posterior margin of gt1 straight but medially with a slight emargination. Gt6 the longest. Syntergum narrower than previous tergite and pointed. Hypopygium large (Fig. 5A). Cercal setae equal. Ovipositor sheaths with visible ventral edge about 4/5 length of hind tibia (Fig. 5A).

### Male
Unknown.

**Etymology**

The generic name (feminine gender) is derived from the Cherangani Hills in Kenya, where the type material was collected.

**Relationships**

*Kerangania* is classified in the subfamily Pteromalinae, tribe Pteromalini based on the following features: (1) antenna with 12 flagellomeres (Fig. 5D); (2) scapulae not anteriorly exposed by pronotum (Fig. 5E); (3) notauli incomplete (Fig. 5E); (4) axillae not strongly advanced (Fig. 5E); (5) axillulae not enlarged (Fig. 5E); (6) marginal vein slender (Fig. 5H); (7) petiole simple (*i.e.*, without anterior flange), very short (Fig. 5G).

*Kerangania* differs from all known Pteromalini genera by the following combination of features: (1) body black, without metallic reflections (Figs. 5A–5G); (2) head anteroposteriorly short (Fig. 5A) (3) clypeal margin bilobed (Fig. 5C); (4) occipital carina present (Fig. 5E); (5) maxillary palpus unusually long and thin (Figs. 5B, 5C and 5F); (6) pronotum separated from lateral lobes of mesoscutum by deep groove; (7) metascutellum extremely short, as a smooth line (Fig. 5G); (8) propodeum short, median area convex, reticulate, with indication of oblique costula (Fig. 5G); (9) petiole extremely short and wide (Fig. 5G); (10) fore wing setation pale, inconspicuous, fringe absent (Fig. 5H); (11) ovipositor sheaths long (Fig. 5A).

In the generic key to Palaearctic Pteromalidae (*Bouček & Rasplus, 1991*) *Kerangania* runs to couplet 162 (*Trichomalopsis* Crawford, 1913 and *Gyrinophagus* Ruschka, 1914) on the account of the distinct occipital carina. *Kerangania* differs from both these genera in virtually all the characters stated above; additionally, it differs from *Gyrinophagus* is having the hind coxa bare and a less stout head. Ignoring the presence of the occipital carina, *Kerangania* would run to couplet 182 and *Lariophagus* Crawford, 1909 on the account of the prominent posterior corners of the propodeum. However, the new genus differs from *Lariophagus* in most features listed above, except for the bilobed clypeus and absent fore wing fringe, the latter character being variable among the species of *Lariophagus*.

In the generic key to Nearctic Pteromalidae (*Bouček & Heydon, 1997*) the new genus also runs to *Trichomalopsis* (couplet 210). Ignoring the occipital carina leads to couplet 232 (*Lariophagus* and *Arthrolytus* Thomson, 1878). In addition to the characters listed above, *Kerangania* also differs from *Arthrolytus* mainly in having the first funicular segment shorter than pedicel (Fig. 5D), the propodeum lacking a median carina or any indication of it (Fig. 5G), and hyaline fore wings (Fig. 5H).

In the generic key to Australasian Pteromalidae (*Bouček, 1988*) *Kerangania* runs to couplet 246 and *Trichomalopsis* (see the discussion above). Ignoring the occipital carina leads to couplets 248–249 (*Canberrana Bouček, 1988*, *Delisleia* Girault, 1936 and *Isoplatoides* Girault, 1913). All of these genera lack most of the diagnostic features of *Kerangania*; additionally, *Canberrana* has the petiole embraced by an extension of the first gastral sternite, which is absent in *Kerangania*.

In the generic key to Oriental Pteromalidae (*Sureshan & Narendran, 2004*) the new genus runs to couplet 91 and *Dibrachys* Förster, 1856 on the account of the absence of the fore wing fringe. *Kerangania* is very different from the latter genus: apart from all of the

features mentioned in the genus diagnosis, it also differs in the position of toruli (lower in *Dibrachys*), fore wing venation (short PMV in *Dibrachys*), and shape of metasoma (shorter and broader in *Dibrachys*). Ignoring the lack of fringe and the presence of the occipital carina leads to couplet 92 and *Trichomalopsis*, and to couplet 96 and *Lariophagus*, respectively (see the discussion above).

The habitus of *Kerangania* has some similarities with *Pteromalus* Swederus, 1795. However, the latter genus lacks most of the diagnostic features of the new genus. *Kerangania* shows no similarities with either of the nine Neotropical genera of Pteromalinae, or the few East Palaearctic or Oriental genera that are not included in any of the above-mentioned keys.

### *Kerangania nuda* Mitroiu, sp. nov.
urn:lsid:zoobank.org:act:5ED1BA05-4E04-4C51-AE48-71F44423B309
(Fig. 5)

**Material examined**

**Holotype**

**KENYA:** ♀, "Kenya: Cherangani Hills, Mt. Chepkotat, 24. vii.1969, From *Lobelia aberdarica*, R. A. Cheke", NHMUK014444241 (NHMUK).

**Paratype**

**KENYA:** 1♀, as holotype, NHMUK014444242 (NHMUK).

**Description**

**Female holotype**
Body length: 3.25 mm. Colour as in Fig. 5. Antenna slightly clavate, clava with microsetation on the third and fourth segments. Striation on lower face almost reaching ventral eye margin, although less extended below toruli. Acropleuron distinctly but shallowly sculptured. Basal cell including basal vein bare. Speculum proximally large and narrowing along the marginal vein. Relative measurements: Head L: 36, W: 79, H: 67; eye H: 38, L: 25; malar space: 20; mouth W: 37; scape L: 32, W 5; pedicel L: 10, W: 5; pedicel plus flagellum L: 63; fu1 L: 7, W: 6; fu6 L: 6, W: 8; clava L: 17, W: 8.5. Mesosoma L: 117, W: 79, H: 75; mesoscutum L: 50, W: 79; mesoscutellum L: 44, W: 50; propodeum L: 22; fore wing L: 260, W: 105; MV: 45; SV: 27; PMV: 40. Metasoma L: 185, W: 70; gt1 L: 30, W 70; gt6 L: 40, W: 40; syntergum L: 20, W: 12.

**Variation**
Body length: 3.25–3.50 mm. Metasomal length 2.6–3.4X maximum width, depending on the degree of tergite retraction during the drying process.

**Etymology**
The name of the species refers to the glabrous appearance of the body and wings (adjective).

## Distribution
Kenya.

## Biology
Both examined specimens have been obtained from *Lobelia aberdarica* R.E.Fr. & T.C.E.Fr. (Campanulaceae), but no other information is available. According to *Plants of the World Online (2023)* *L. aberdarica* is native to Kenya and Uganda.

### *Pilosalis* Mitroiu, Rasplus & van Noort, gen. nov.
urn:lsid:zoobank.org:act:F2772385-9073-45D7-BA3C-3B2AA0791E62
(Figs. 6–11)

## Type species
*Pilosalis barbatulus* Mitroiu, sp. nov., here designated.

## Diagnosis

### Both sexes
Head and mesosoma with dense setation (Figs. 6B, 6E, 6F, 7B, 7C, 8B, 8E, 8F, 9B, 9E, 9F; 10B, 10E, 10F, 11B, 11E and 11F); head long anteroposteriorly, especially in males (Figs. 6A, 7C, 8A, 9A and 11A); eyes very large, consequently malar space very short (Figs. 6B, 7C, 8B, 9B, 10B and 11B); gena with large hollow at mouth margin (Figs. 6F, 9B, 11B and 11C); mandibles very large, falcate (Figs. 6B, 11B and 11C); toruli at least slightly above center of face, usually much higher (Figs. 6B, 8B, 9B, 10B and 11B); antenna 11354 (Figs. 6D, 8D, 9D, 10B; 11D); occipital carina absent (Figs. 6E, 9E and 11E); mesosoma short (Figs. 6F, 8F, 9F, 10D and 11F); notauli almost absent, restricted to basal pits (Figs. 6E, 8E, 9E, 10C and 11E); frenal area distinct (Figs. 6E and 11E); propodeum without carinae or nucha (Figs. 6G, 8G, 9G, 10E and 11G); fore wing entirely setose, with wide costal cell (Figs. 6H, 8H, 9H, 10F and 11H); petiole smooth, long conical but flattened, without anterior flange but with short lamina reaching nucha, ventrally embraced by short extensions of gs1 (Figs. 6G, 8G, 9G, 10E and 11G).

## Description

### Female
Body fairly robust, with at least slight metallic reflections (Figs. 6–11). Head and dorsal side of mesosoma mostly with short dense setation (Figs. 6B, 6E, 6F, 8B, 8E, 8F, 9B, 9E, 9F, 10B, 10E, 10F, 11B, 11E and 11F), longer on mesoscutellum in all species (Figs. 6E, 9E and 11E), and on lower part of the head in *P. barbatulus* sp. nov. (Fig. 6C).

Head wider than high in frontal view, and long anteroposteriorly, temples large; vertex often strongly arched (Figs. 6B, 8B and 11B). Clypeus reticulate, clypeal margin symmetric, slightly arched (Fig. 11C) or with broad triangular projection, which may be difficult to see being slightly curved inwards and sometimes obscured by setae (Figs. 6C, 8C, 9C and 10B). Lower face on each side of clypeus with more or less developed blade-like projection delimiting the anterior margin of the large malar depression (Figs. 6B, 8B, 9B, 10B and 11B). Tentorial pits absent. Scrobal depression deep, with large raised triangle separating

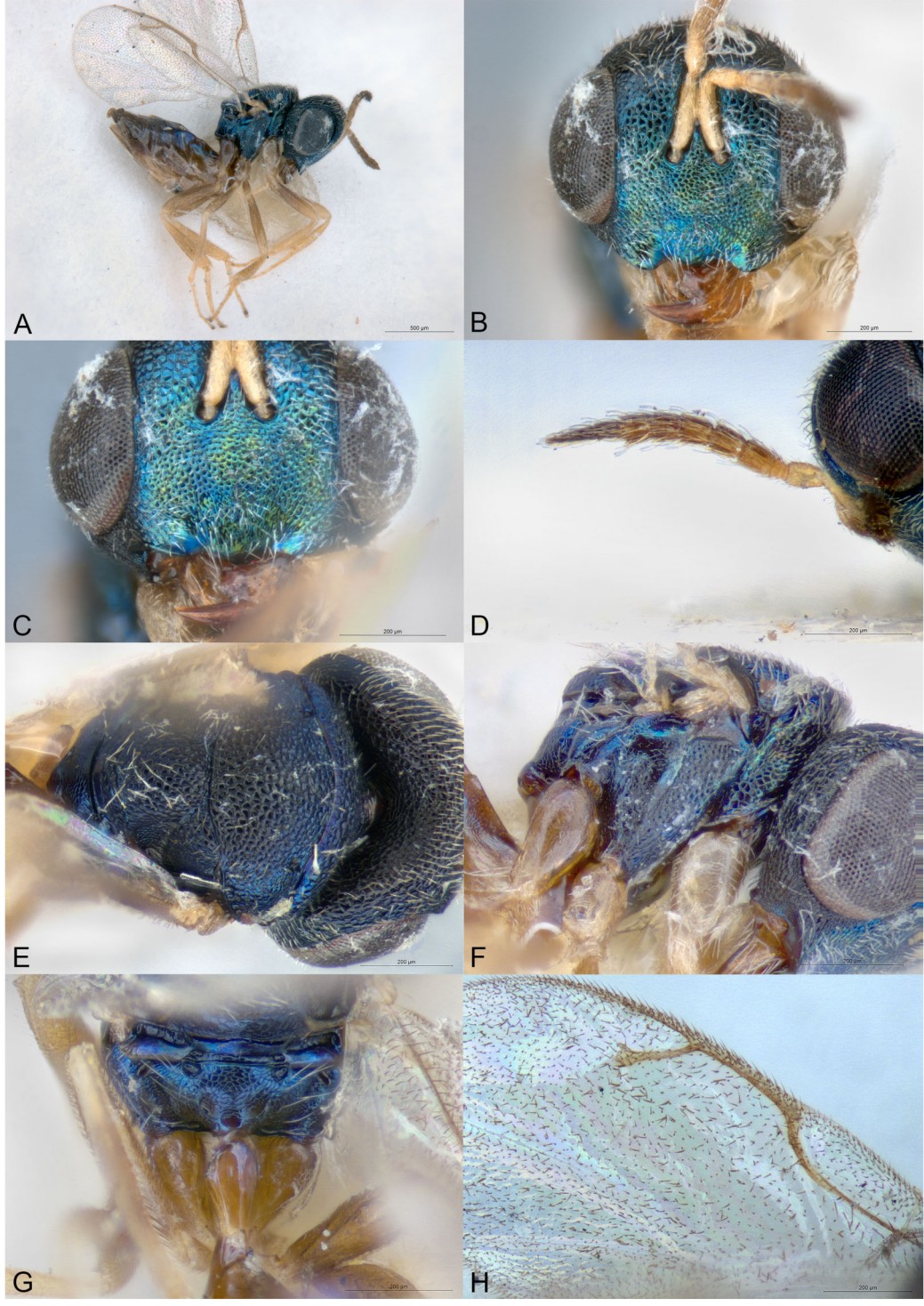

**Figure 6** *Pilosalis barbatulus.* (A) Female holotype, habitus, lateral. (B) Female holotype, head, frontal. (C) Female holotype, clypeus. (D) Female paratype, antenna, lateral. (E) Female holotype, mesosoma, dorsal. (F) Female holotype, mesosoma, lateral. (G) Female paratype, propodeum, dorsal. (H) Female holotype, fore wing, dorsal.

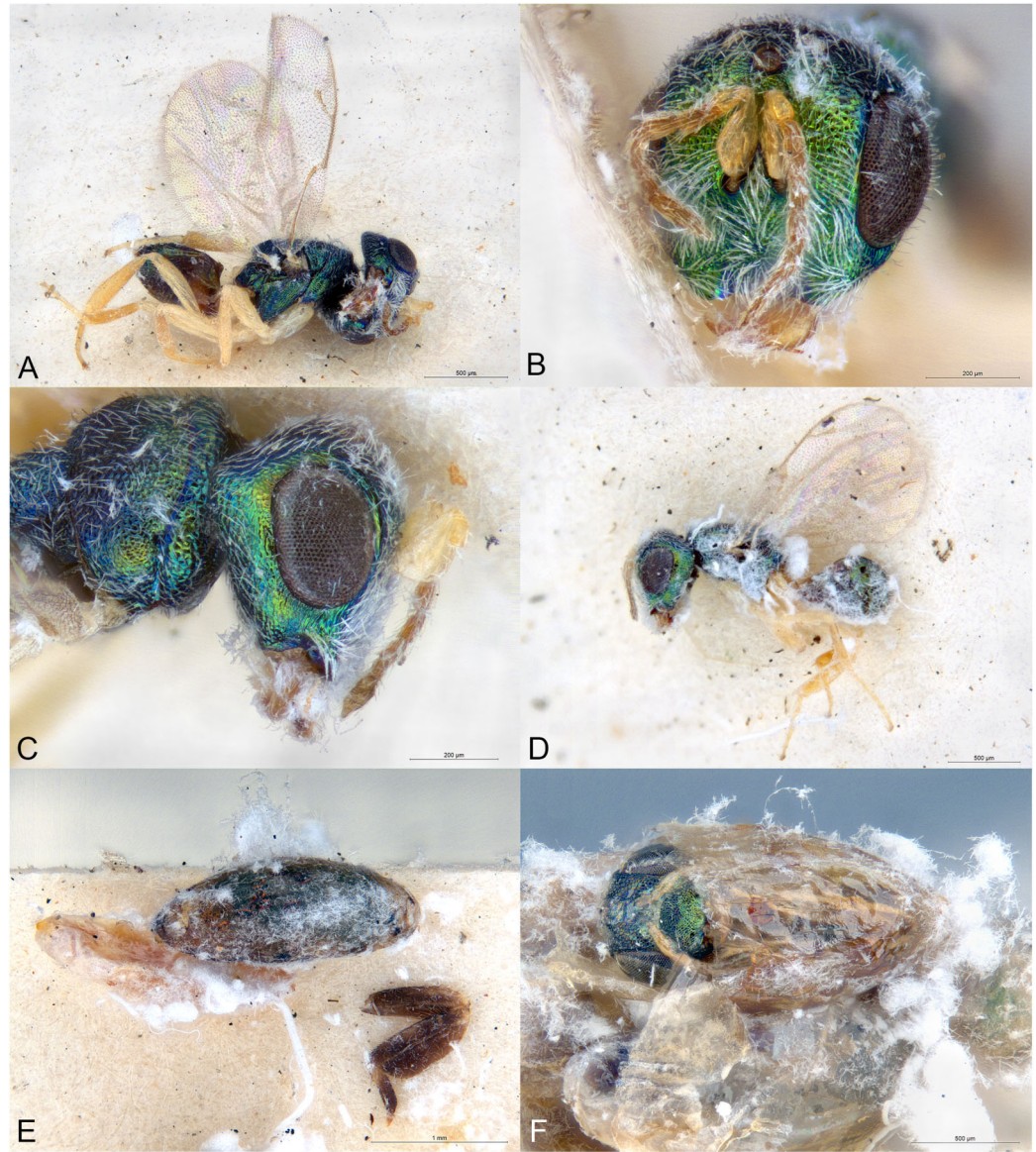

**Figure 7** *Pilosalis barbatulus.* (A) Allotype male, habitus, lateral. (B) Paratype male, head, frontal. (C) Allotype male, head, lateral. (D) Male covered in host filaments. (E) Parasitized host and presumed host leg. (F) Parasitoid inside host remains.

toruli (Figs. 6B, 8B, 9B; 10B and 11B). Gena hollowed at mouth corner (Figs. 6F, 9B, 11B and 11C). Genal carina absent. Malar sulcus present or absent. Eyes large, their inner margin virtually parallel or converging in lower part (Figs. 6B, 8B, 9B, 10B and 11B). Occiput usually strongly concave, without carina (Figs. 6E, 9E and 11E). Antennal insertion at least slightly above center of face, usually much higher (Figs. 6B, 8B, 9B, 10B and 11B). Antennal formula 11354 (Figs. 6D, 8D, 9D, 10B and 11D). Anelli strongly transverse. Antennal scape short, normal (Figs. 6B, 8B, 9B and 10B) or with large ventral lamina (Fig. 11D). Antennal clava symmetric, without conspicuous area of microsetation,

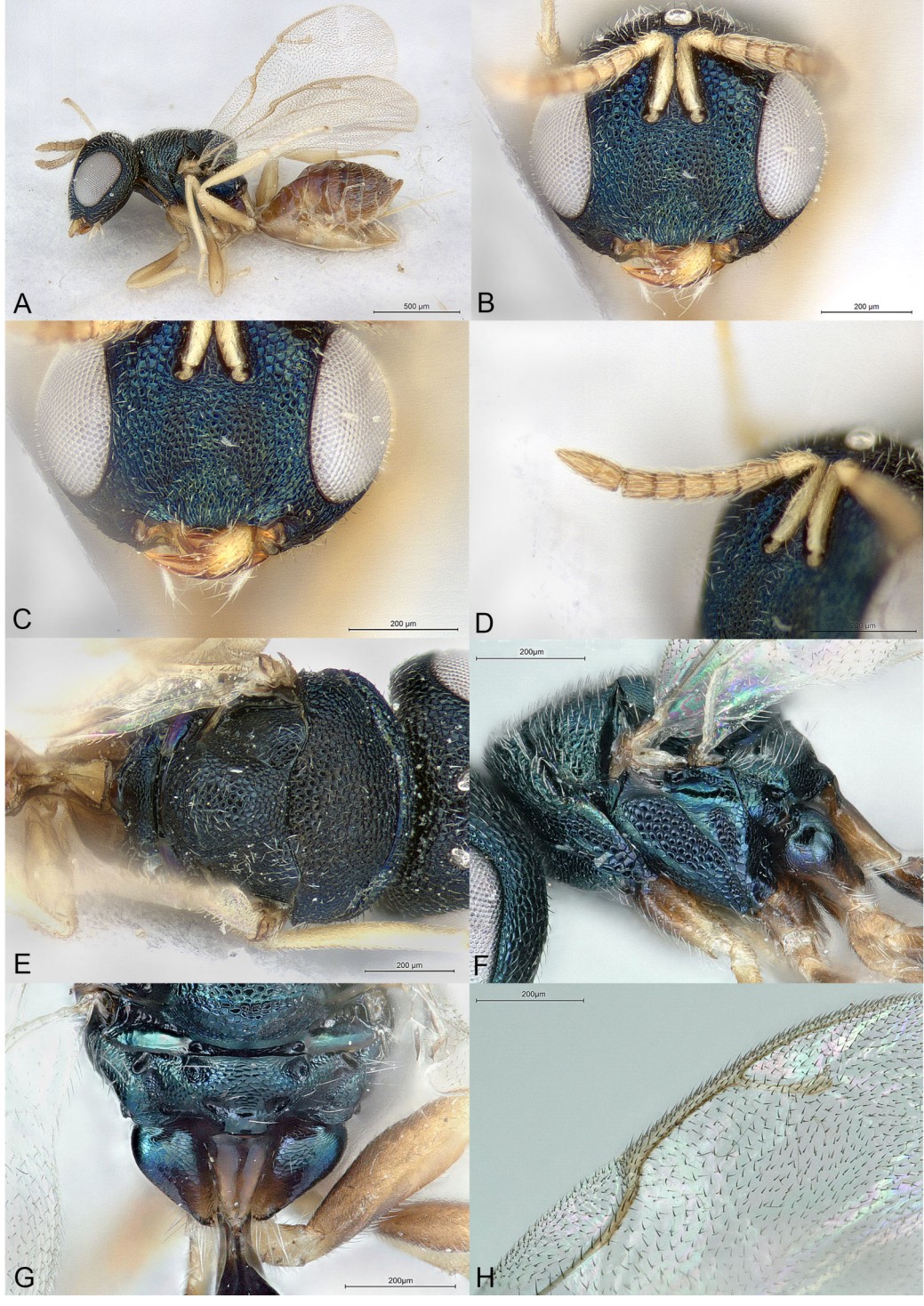

**Figure 8** *Pilosalis bouceki.* (A) Female holotype, habitus, lateral. (B) Female holotype, head, frontal. (C) Female holotype, clypeus. (D) Female holotype, antenna, lateral. (E) Female holotype, mesosoma, dorsal. (F) Female paratype, mesosoma, lateral. (G) Female paratype, propodeum, dorsal. (H) Female paratype, fore wing, dorsal.

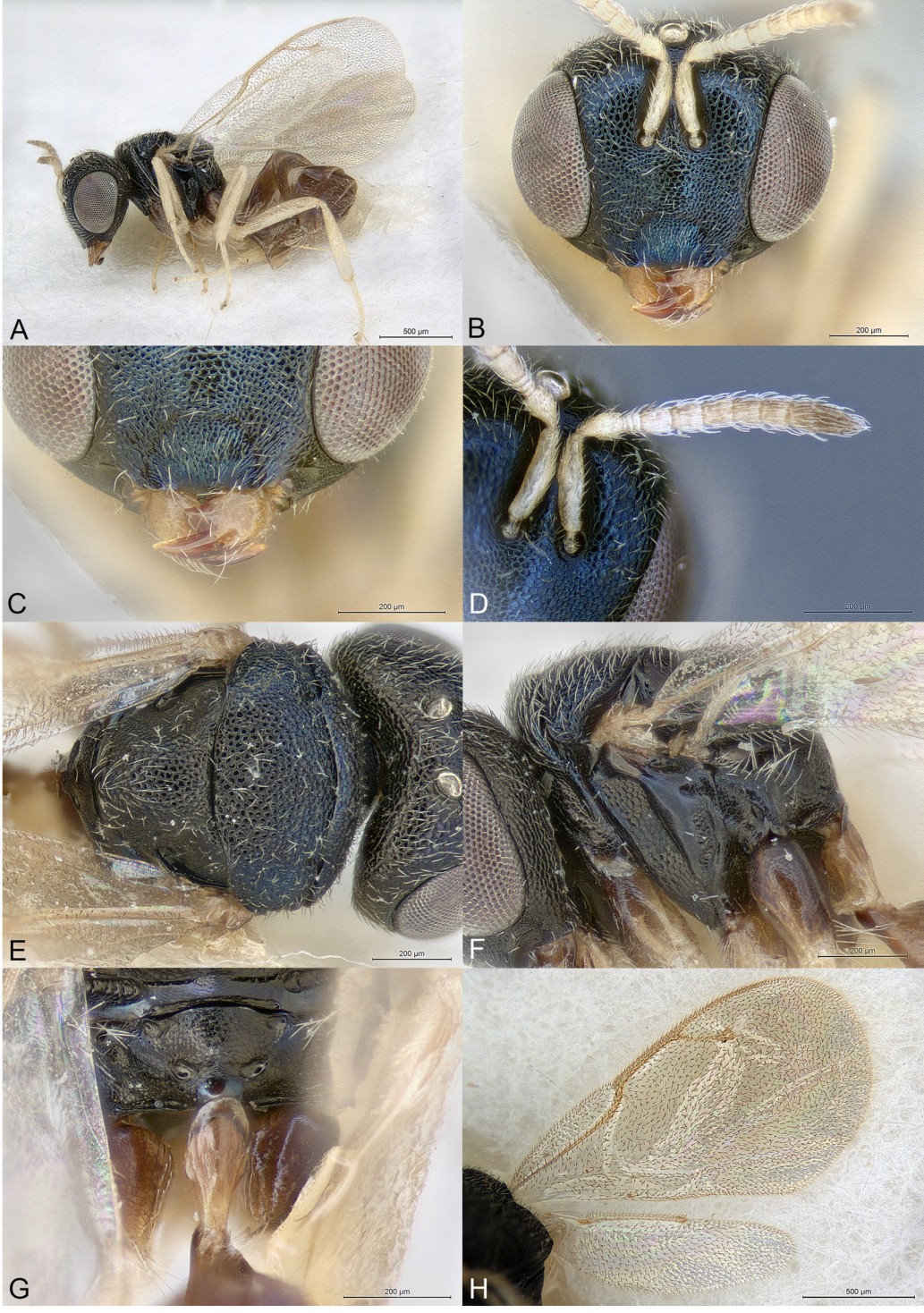

**Figure 9** *Pilosalis eurys.* (A) Female holotype, habitus, lateral. (B) Female holotype, head, frontal. (C) Female holotype, clypeus. (D) Female holotype, antenna, lateral. (E) Female holotype, mesosoma, dorsal. (F) Female paratype, mesosoma, lateral. (G) Female paratype, propodeum, dorsal. (H) Female paratype, fore and hind wings, dorsal.

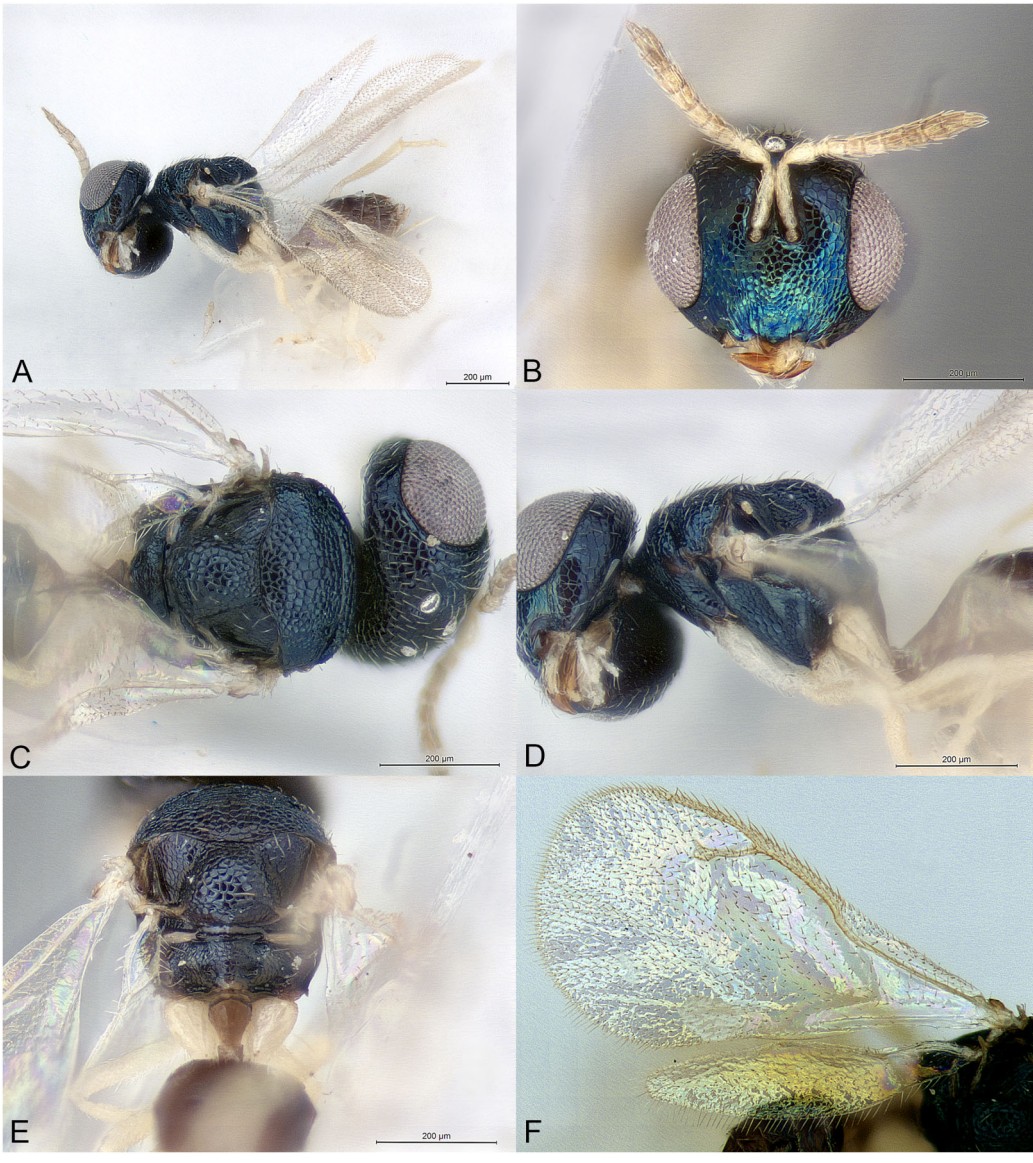

**Figure 10 *Pilosalis minutus*.** (A) Female holotype, habitus, lateral. (B) Female holotype, head including antennae, frontal. (C) Female holotype, mesosoma, dorsal. (D) Female holotype, mesosoma, lateral. (E) Female paratype, propodeum, dorsal. (F) Female holotype, fore and hind wings, dorsal.

distal end acuminate but not pointed (Figs 6D, 8D, 9D, 10B, 11D). Mandibles very large, falcate (Figs. 6B, 11B and 11C).

Mesosoma convex (Figs. 6F, 8F, 9F, 10D and 11F). Pronotum short conical, almost as broad as mesoscutum (Figs. 6E, 8E, 9E, 10C and 11E). Pronotal collar with anterior margin rounded, not carinated. Mesoscutum very short. Notauli indicated as round pits at anterior margin of mesoscutum (Figs. 6E, 8E, 9E, 10C and 11E). Axillae slightly advanced. Mesoscutellum convex. Frenal line absent, but frenal area with coarser reticulation than rest of mesoscutellum (Figs. 8G, 10E and 11G). Metascutellum short. Propodeum short

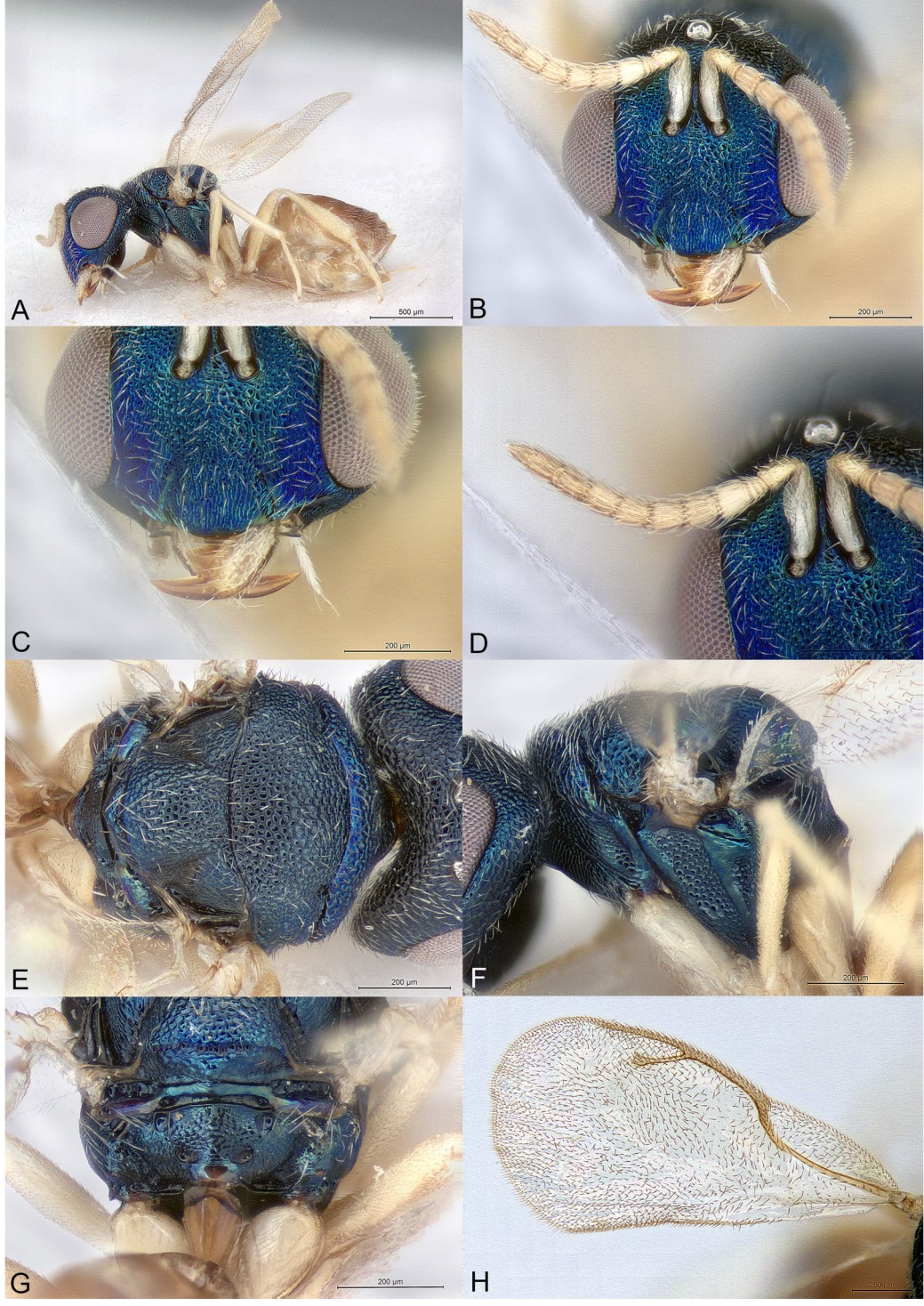

**Figure 11 *Pilosalis platyscapus.*** (A) Female holotype, habitus, lateral. (B) Female holotype, head, frontal. (C) Female holotype, clypeus. (D) Female holotype, antenna, lateral. (E) Female holotype, mesosoma, dorsal. (F) Female holotype, mesosoma, lateral. (G) Female holotype, propodeum, dorsal. (H) Female holotype, fore wing, dorsal.               

(Figs. 6G, 8G, 9G, 10E and 11G). Median area convex, reticulate except four small round foveae, two near the anterior margin of propodeum and two on sides of nuchal area. Plicae, median carina and costula absent. Nucha indicated only as a small smooth area. Propodeal hind corners not prominent or sharp. Propodeal spiracles large oval, at the anterior margin of propodeum. Prepectus about as long as high, with shallow uniform reticulation. Mesopleuron with at least the acropleuron smooth, without any pits, ventrally without transverse carina (Figs. 6F, 8F, 9F, 10D and 11F). Hind coxa long conical, dorsally bare. Hind tibia with one spur (the second greatly reduced). Wings hyaline (Figs. 6H, 7A, 7D, 8H, 9H, 10F and 11H). Fore wing completely setose. Costal cell wide, length about 4.4–5.4X width. Venation not widened. Parastigma without hyaline break. Stigmal vein considerably shorter than both marginal and postmarginal veins. Stigma moderately capitate.

Metasoma petiolate. Gaster oval. Petiole longer than broad, conical, flattened dorso-ventrally, smooth; in anterior part wider and with thin lamina touching or virtually touching nucha, but without anterior flange; in posterior part ventrally embraced by short extensions of gs1 (Figs. 6G, 8G, 9G, 10E and 11G). All gastral tergites normal, not enlarged. Hypopygium large, extending at least beyond middle of gaster (Figs. 8A and 11A). Cercal setae equal. Ovipositor sheaths short (Figs. 6A, 8A, 9A, 10A and 11A).

**Male**
Known only for *P. barbatulus* sp. nov. and similar to the female in most features, except the antennal scape which is ventrally expanded into a large lamina (Fig. 7B) as in the *P. platyscapus* sp. nov. female, and the temples, which are distinctly enlarged as compared to the female's (Fig. 7C). The colour of the head, mesosoma and legs is also lighter than in females (Figs. 7A–7D).

**Etymology**
The generic name (masculine gender) refers to the entirely setose wings.

**Relationships**
*Pilosalis* is classified in the subfamily Pteromalinae, tribe Pteromalini, based on the following features: (1) antenna with 12 flagellomeres (Figs. 6D, 8D, 9D, 10B and 11D); (2) scapulae not anteriorly exposed by pronotum (Figs. 6E, 8E, 9E, 10C and 11E); (3) notauli incomplete (Figs. 6E, 8E, 9E, 10C and 11E); (4) axillae not strongly advanced (Figs. 6E, 8E, 9E, 10C and 11E); (5) axillulae not enlarged (Figs. 6E, 8E, 9E, 10C and 11E); (6) marginal vein slender (Figs. 6H, 7A, 7D, 8H, 9H, 10F and 11H); (7) petiole simple, *i.e.*, without anterior flange (the anterior lamina must not be confused with the anterior flange), and long (Figs. 6G, 8G, 9G, 10E and 11G).

*Pilosalis* differs from all known genera of Pteromalini in having the the following combination of features: (1) head and mesosoma with dense setation (Figs. 6B, 6E, 6F, 7B, 7C, 8B, 8E, 8F, 9B, 9E, 9F, 10B, 10E, 10F, 11B, 11E and 11F); (2) head long anteroposteriorly (Figs. 6A, 7C, 8A, 9A and 11A); (3) eyes large hence malar space short (Figs. 6B, 7C, 8B, 9B, 10B and 11B); (4) mandibles very large, falcate hence gena with large hollow at mouth margin (Figs. 6B, 11B and 11C); (5) fore wing wide and entirely setose

(Figs. 6H, 7A, 7D, 8H, 9H, 10F and 11H); (6) petiole long conical but flattened dorso-ventrally, smooth, ventrally embraced by short extensions of gs1 and with anterior lamina (Figs. 6G, 8G, 9G, 10E and 11G).

In the generic key to Palaearctic Pteromalidae (Bouček & Rasplus, 1991) *Pilosalis* runs to couplet 83 and *Panstenon* Walker, 1846 based on the high toruli, but the new genus differs from *Panstenon* in almost all important features. If the position of the toruli is ignored, of the following genera, *Pilosalis* shares with *Cratomus* Dalman, 1820 and *Paracarotomus* Ashmead, 1894 an unusually long head anteroposteriorly, but greatly differs from both genera in many details of head and mesosoma structure. Another genus with an elongated petiole that is ventrally embraced by extensions of gs1 is *Toxeumorpha* Girault, 1915. *Pilosalis* differs from *Toxeumorpha* in having a different head shape, a much shorter mesosoma, a different structure of the petiole, and entirely setose wings among many other features.

In the generic key to Nearctic Pteromalidae (Bouček & Heydon, 1997) the new genus runs to couplet 131 and *Vrestovia* Bouček, 1961 on the account of the lack of propodeal carinae. However, other features of *Vrestovia* are very different, such as the carinated pronotal collar, the shape of the anelli, and the colour of the mesosomal setae, to mention only those given in the key. Choosing the opposite part of the couplet leads to *Propodeia* Bouček, 1993 and *Heteroschema* Gahan, 1919. Of these, only the first genus has both a large hollow at mouth corner and the gs1 provided with a distinct flange; *Pilosalis* differs from *Propodeia* in many features, especially the shape of clypeus (with two broad teeth in *Propodeia*), the propodeal structure (with plicae, median carina, costula and nucha in *Propodeia*), and the shape and sculpture of the petiole (rugose, much shorter and with parallel sides in *Propodeia*). Of the Neotropical species that are keyed here, *Pilosalis* most closely resembles *Toxeumella* Girault, 1913 mostly due to its head shape, falcate mandibles and dense body setation. However, the new genus differs from *Toxeumella* mostly as follows: (1) eyes larger (normal in *Toxeumella*); (2) occipital carina absent (present in *Toxeumella*); (3) clava without any strip of microsetation (with long strip in *Toxeumella*); (4) notauli incomplete, restricted to anterior pits (complete in *Toxeumella*); (5) propodeum without plicae or carinae (with plicae, costula and short median carina in *Toxeumella*); (6) gaster petiolate (gaster sessile in *Toxeumella*).

In the generic key to Australasian Pteromalidae (Bouček, 1988) *Pilosalis* runs to couplet 278 and *Yanchepia* Bouček, 1988, although the clypeal margin is not exactly as described; also, in *Pilosalis* the pronotal collar is not carinate and the petiole is longer than propodeum. Other differences from *Yanchepia* include: (1) no occipital carina; (2) tentorial pits inconspicuous; (3) gena much shorter; (4) toruli considerably higher; (5) antenna 11354; (6) clava without large microsetation area; (7) apex of scutellum without small upturned median tooth; (8) propodeum without nucha, median carina and plicae; (9) petiole much longer than broad. Of the Australasian genera *Pilosalis* is also superficially similar with *Acroclisella* Girault, 1915 and *Laticlypa* Bouček, 1988 in the general head shape and the large mandibles; however, the new genus greatly differs from both these genera in many features. It also greatly differs from *Trigonogastrella* Girault, 1915 where the petiole also has an anterior lamina (Bouček, 1988).

In the generic key to Oriental Pteromalidae (*Sureshan & Narendran, 2004*) the new genus runs to couplet 48 and *Narendrella Sureshan, 1999* on the account of the high toruli, antennal formula and dense setation (although generally very short in *Pilosalis*). *Pilosalis* differs from the latter genus in many characters, mainly the shape of head, including clypeus, the structure of propodeum and petiole, and the fore wing venation and setation pattern (*Sureshan, 1999*).

*Pilosalis* has no strong similarities with either of the nine Neotropical genera of Pteromalinae, or the few East Palaearctic or Oriental genera that are not included in any of the above-mentioned keys.

**Key to *Pilosalis* species (females)**

1 Scape with strong ventral lamina (Fig. 11D); clypeal margin without median triangular projection (Fig. 11C); malar sulcus present; MV 1.8-1.9X SV; face bright blue-violet to blue-green (Fig. 11C)… **P. platyscapus Mitroiu, Rasplus & van Noort, sp. nov.**

- Scape without ventral lamina (Figs. 6B, 8B, 9B and 10B); clypeal margin with blunt median triangular projection, which may be difficult to see being slightly curved inwards and sometimes obscured by setae (Figs. 6C, 8C, 9C and 10B); malar sulcus absent; MV 1.45-1.70X SV; face sometimes darker (Figs. 8C and 9B) … 2.

2 (1) Lateral side of mesosoma blackish (Fig. 9F); mesepimeron with deep narrow rugose-reticulate depression towards posterior margin, the surrounding areas smooth (Fig. 9F); legs except coxae pale yellow (Fig. 9A) … **P. eurys Mitroiu & van Noort, sp. nov.**

- Lateral side of mesosoma with strong bluish reflections (Figs. 6F, 8F and 10D); mesepimeron with shallower and larger reticulate depression in the middle, the surrounding areas at least delicately reticulate (Figs. 6F, 8F and 10D); legs sometimes darker (Figs. 6A and 8A) … 3.

3 (2) Lower face with paraclypeal lobes very large (Figs. 6B and 6C); clypeal setae long, conspicuous (Fig. 6C) … **P. barbatulus Mitroiu, sp. nov.**

- Lower face with paraclypeal lobes smaller (Figs. 8B, 8C and 10B); clypeal setae short, hardly visible (Figs. 8C and 10B) … 4.

4 (3) Hypopygium not reaching tip of gaster; legs pale yellow except white fore coxae (Fig. 10A); flagellum pale yellow, distal part becoming brownish (Fig. 10B) … **P. minutus Mitroiu, sp. nov.**

- Hypopygium virtually reaching tip of gaster (Fig. 8A); legs extensively brownish, especially basally (Fig. 8A); flagellum light brown (Fig. 8D) … **P. bouceki Mitroiu & Rasplus, sp. nov.**

**Pilosalis barbatulus Mitroiu, sp. nov.**
urn:lsid:zoobank.org:act:82A2EAA5-A2D1-4C1A-9DB4-1331F3DCCBA7
(Figs. 6, 7)

**Material examined**

**Holotype**

 

GHANA: ♀, "Gold Coast, Aburi. 31.I.1922, W.H. Patterson", "Ex. *Ptyelus grossus*, F.", NHMUK014444243 (NHMUK).

**Allotype**

GHANA: ♂, as holotype, NHMUK014444244 (NHMUK).

**Additional paratypes**

GHANA: 22♀, as holotype, NHMUK014444246 to NHMUK014444267 (NHMUK); 1♀, as holotype, MICO-2023-3; 1♂, as holotype, NHMUK014444269 (NHMUK); 1♂, as holotype, MICO-2023-4 (MICO).

**Additional material**

GHANA: 12♀, 3♂, as holotype, NHMUK014444270 to NHMUK014444284 (NHMUK).

**Description**

**Female holotype**
Body length: 1.75 mm. Colour as in Fig. 6. Clypeal margin with median triangular projection, which may be difficult to see because it is slightly curved inwards and sometimes obscured by setae (Fig. 6C). Paraclypeal lobes very large, these and clypeus covered by long white setae (Figs. 6B and 6C). Malar sulcus absent. Scape without ventral lamina (Fig. 6B). Upper mesepimeron very delicately reticulate, appearing almost smooth (Fig. 6F). Hypopygium reaching tip of gaster (Fig. 6A). Relative measurements: Head L: 34, W: 62, H: 50; eye H: 35, L: 27; malar space: 8; mouth W: 26; scape L: 16, W: 4; pedicel L: 7, W: 4; pedicel plus flagellum L: 50; fu1 L: 6, W: 4; fu5 L: 5, W: 4; clava L: 14, W: 5.5. Mesosoma L: 61, W: 50, H: 48; mesoscutum L: 25, W: 50; mesoscutellum L: 22, W: 22; propodeum L: 15; fore wing L: 130, W: 65; MV: 25; SV: 15; PMV: 31 (distal end difficult to set). Metasoma. Petiole L: 20, W: 10; gaster L: 95, W: 20.

**Male allotype**
Differs from the female holotype mainly as follows. Body length: 1.5 mm. Coloration of head, mesosoma and legs lighter (Figs. 7A–7C). Temple much larger, conspicuously inflated behind eye (Fig. 7C). Scape ventrally expanded into a distinct lamina (Fig. 7B). MV about 1.8X SV. Gaster much shorter (Figs. 7A and 7D).

**Variation**

**Female**
Body length: 1.75–1.85 mm. MV 1.5–1.7X SV. Petiole length 1.8–2.0X width. Gaster size variable depending on its collapse degree (occasionally strongly compressed laterally). The specimens excluded from the type series are either almost entirely covered in white secretions, or damaged so their features are difficult or impossible to examine; they definitely belong to the same species but could not be measured and included in the above stated variation.

**Male**

Body length: 1.5–1.7 mm.

**Etymology**

The specific epithet (adjective) refers to the unusual long facial setae (from the Latin *barbatulus* meaning "with a little beard").

**Distribution**

Ghana.

**Biology**

All examined specimens have been labeled "Ex. *Ptyelus grossus*, F." However, three additional cards also bearing this label have several host remains that suggest a different host. These host remains (some still with parasitoids inside, Figs. 7E and 7F) are ovoid sac-like structures (? mummies) covered in white waxy filaments identical to those found on many of the above specimens (Fig. 7D). This suggests that the hosts are most probably mealybugs (Hemiptera: Pseudococcidae) or related hemipterans and not *P. grossus* (which produces foam and not waxy filaments). Further evidence is one leg found together with the host remains, which generally resembles mealybug legs *i.e.*, it has one tarsal segment (Fig. 7E). Thus, the ovoid cocoon-like structures are probably mummies, *i.e.*, parasitized nymphs or females of an unknown mealybug. Other species of *Pilosalis* are expected to have similar hosts. Interestingly, the genera *Austroterobia* Girault, 1938 and *Teasienna* Heydon, 2004 (Pteromalidae: Pachyneurinae), which parasitize giant scales (Hemiptera: Coccoidea: Monophlebidae), have some superficial similarities with *Pilosalis*, such as falcate mandibles and entirely setose fore wings (*Mitroiu, 2017*).

***Pilosalis bouceki*** Mitroiu & Rasplus, sp. nov.

urn:lsid:zoobank.org:act:0902BB2C-AE87-4780-99EB-2EED1BAF1BB5

(Fig. 8)

**Material examined**

**Holotype**

**ZIMBABWE:** ♀, "Zimbabwe, nr. Harare, ii.1981, A. Watsham", NHMUK014444285 (NHMUK).

**Paratypes**

**KENYA:** 1♀, "Kenya, Rift Valley Prov., Matthews Range, 1,459 m, 0.97984°N, 37.34599°E", "Malaise trap, riverine forest, near Wamba, 3–17 MAY 2016, R. Copeland", JRAS08824_0101 (CBGP). **SOUTH AFRICA:** 1♀, "South Africa, Nylsvley Res, Tvl. ii.1979, M. W. Mansell", "By sweeping", "National Coll. of Insects Pretoria, S. Afr.", 27,493 (NMPC); 1♀, "South Africa: NW, Farm Mezeg, Enzelsberg, 20 km NE of Zeerust", "25.22S 26.13E 1,200 m, 25.iii.1996 R. Urban", "National Coll. of Insects Pretoria, S. Afr.", 27,494 (NMPC). **ZIMBABWE:** 1♀, as holotype, NHMUK014444286 (NHMUK); 2♀,

"Zimbabwe, Salisbury, Jan. 81, A. Watsham", NHMUK014444287, NHMUK014444288 (NHMUK); 1♀ "Zimbabwe, Salisbury, vii. 1978, A. Watsham", NHMUK014444289 (NHMUK).

**Description**

**Female holotype**

Body length: 1.75 mm. Colour as in Fig. 8. Clypeal margin with median triangular projection, which may be difficult to see because it is slightly curved inwards and sometimes obscured by setae (Fig. 8C). Paraclypeal lobes small, these and clypeus covered by short setae (Figs. 8B and 8C). Malar sulcus absent. Scape without ventral lamina (Figs. 8B and 8D). Upper mesepimeron virtually smooth (Fig. 8F). Hypopygium reaching tip of gaster (Fig. 8A). Relative measurements: Head L: 32, W: 60, H: 51; eye H: 31, L: 22; malar space: 8; mouth W: 25; scape L: 16, W: 3; pedicel L: 8, W: 4; pedicel plus flagellum L: 47; fu1 L: 5, W: 4.5; fu5 L: 5, W: 4.5; clava L: 12, W: 5. Mesosoma L: 60, W: 53, H: 47; mesoscutum L: 25, W: 53; mesoscutellum L: 25, W: 25; propodeum L: 14; fore wing L: 120, W: 63; MV: 24; SV: 15.5; PMV: 30 (distal end difficult to set). Metasoma. Petiole L: 20, W: 10; gaster L: 77, W: 35.

**Male**

Unknown.

**Variation**

Body length: 1.75–2.00 mm. MV 1.5–1.7X SV. Petiole length 2.0–2.2X width. Gaster size variable depending on its collapse degree (occasionally strongly compressed laterally).

**Etymology**

The specific epithet is dedicated to Zdenek Bouček, who first acknowledged this genus (noun in genitive case).

**Distribution**

Kenya, South Africa, Zimbabwe.

**Biology**

Unknown.

**_Pilosalis eurys_ Mitroiu & van Noort, sp. nov.**

urn:lsid:zoobank.org:act:D3A6FA39-2003-40D5-8E19-27639D6AAD33

   (Fig. 9)

**Material examined**

**Holotype**

**CAMEROON:** ♀, "Cameroon: Nkoemvon, VIII.1978, D. Jackson", "♀ Pilosalis eurys", 27,495 (NMPC).

**Paratypes**

**CAMEROON:** 1♀, as holotype, 27,496 (NMPC). **CENTRAL AFRICAN REPUBLIC:** 1♀, "Central African Republic, Prefecture Sangha-Mbaéré, Parc National de Dzanga-Ndoki, Mabéa Bai, 21.4 km 53° NE Bayanga", "3°02.01′N 16°24.57′E, 510 m, 1–7.v.2001, S. van Noort, Yellow pan, CAR01-Y18, Lowland Rainforest, marsh clearing", SAM-HYM-P078965 (SAMC); 1♀, as previous, SAM-HYM-P082130 (SAMC); 1♀, "Central African Republic, Prefecture Sangha-Mbaéré, Réserve Spéciale de Forêt Dense de Dzanga-Sangha, 12.7 km 326° NW Bayanga", "3°00.27′N 16°11.55′E, 420 m, 11–17.v.2001, S. van Noort, Yellow pan, CAR01-Y28, Lowland Rainforest", SAM-HYM-P078970 (SAMC); 11♀, as previous, CAR01-Y28, CAR01-Y34, CAR01-Y38, CAR01-Y40, CAR01-Y43, CAR01-Y50, SAM-HYM-P078966 to SAM-HYM-P078969, SAM-HYM-P082126, SAM-HYM-P082127, SAM-HYM-P082131, SAM-HYM-P082580, SAM-HYM-P082581; SAM-HYM-P082583, SAM-HYM-P082584 (SAMC).

**Description**

**Female holotype**
Body length: 1.9 mm. Colour as in Fig. 9. Clypeal margin with median triangular projection, which may be difficult to see because it is slightly curved inwards and sometimes obscured by setae (Fig. 9C). Paraclypeal lobes small, these and clypeus covered by short setae (Figs. 9B and 9C). Malar sulcus absent. Scape without ventral lamina (Figs. 9B and 9D). Most part of mesepimeron smooth, with only an oval reticulate depression towards posterior margin (Fig. 9F). Hypopygium reaching beyond middle of gaster (Fig. 9A). Relative measurements: Head L: 39, W: 73; H: 56; eye H: 39, L: 28; malar space: 8; mouth W: 15; scape L: 16, W: 4; pedicel L: 8, W: 4.5; pedicel plus flagellum L: 50; fu1 L: 5, W: 4.5; fu5 L: 5, W: 5; clava L: 13, W: 5.5. Mesosoma L: 70, W: 63, H: 55; mesoscutum L: 29, W: 63; mesoscutellum L: 27, W: 30; propodeum L: 15; fore wing L: 147, W: 80; MV: 32; SV: 22; PMV: 40. Metasoma. Petiole L: 22, W: 11; gaster L: 80, W: 50.

**Male**
Unknown.

**Variation**
Body length: 1.9–2.1 mm. MV 1.45–1.70X SV. Gaster size variable depending on its collapse degree (occasionally strongly compressed laterally).

**Etymology**
The name of the species (adjective) was Bouček's choice (see label of holotype) and probably refers to the wide fore wing (and costal cell) that is characteristic for *Pilosalis* species.

**Distribution**
Cameroon, Central African Republic.

## Biology

Unknown.

## *Pilosalis minutus* Mitroiu, sp. nov.

urn:lsid:zoobank.org:act:6259AE7E-AED9-4574-97A0-2A77B7E99565

(Fig. 10)

## Material examined

### Holotype

**CAMEROON:** ♀, "Cameroon: Douala, *Elaeis guineensis* palm trees, *Chromolaena odorata etc.*, IV–V.2010, Mal. Tr., Kekenou S.", NHMUK014444290 (NHMUK).

## Description

### Female holotype
Body length: 0.95 mm. Colour as in Fig. 10. Clypeal margin with median triangular projection (Fig. 10B). Paraclypeal lobes small, these and clypeus covered by short setae (Fig. 10B). Malar sulcus absent. Scape without ventral lamina (Fig. 10B). Upper mesepimeron finely reticulate (Fig. 10D). Hypopygium reaching about 3/4 of gaster length. Relative measurements: Head L: 26, W: 48, H: 40; eye H: 25, L: 21; malar space: 8; mouth W: 20; scape L: 13, W: 35; pedicel L: 7, W: 4; pedicel plus flagellum L: 41; fu1 L: 3.5, W: 3.5; fu5 L: 4.5, W: 4.5; clava L: 12, W: 5. Mesosoma L: 48, W: 39, H: 34; mesoscutum L: 17, W: 39; mesoscutellum L: 17, W: 17; propodeum L: 10; fore wing L: 110, W: 55; MV: 23; SV: 14; PMV: 25. Metasoma. Petiole L: 14, W: 8; gaster L: 50, W: 30.

### Male
Unknown.

### Etymology
The name of the species refers to small size of the holotype (adjective).

### Distribution
Cameroon.

### Biology
Unknown.

## *Pilosalis platyscapus* Mitroiu, Rasplus & van Noort, sp. nov.

urn:lsid:zoobank.org:act:1B2260C7-A006-4800-B62A-4D8F0270B6A2

(Fig. 11)

## Material examined

### Holotype

**CAMEROON:** ♀, "Nkoemvon, 13.vii-24.viii.1980, D. Jackson", NHMUK014444291 (NHMUK).

**Paratypes**

**GABON:** 1♀, "Gabon, Prov. Ogoové-Maritime, Réserve de la Moukalaba-Dougoua, 12.2 km 305° NW Doussala, 2°17.00′S 10°29.83′E, 110 m", "24–25.ii.2000, S. van Noort, Malaise trap, GA00-M03, Coastal Lowland Rainforest, forest margin in large clearing", SAM-HYM-P0023796 (SAMC). **KENYA:** 1♀, "Kenya, Eastern Prov., Endau Mtn., base of, 531 m, 1.30026°S, 38.52805°E″, "Malaise trap, in indigenous forest, 25 JAN–8 FEB 2016, R. Copeland", JRAS08825_0101 (CBGP).

**Description**

**Female holotype**
Body length: 1.8 mm. Colour as in Fig. 11. Clypeal margin without median triangular projection, slightly and almost evenly curved (Fig. 11C). Paraclypeal lobes small, these and clypeus covered by short setae (Figs. 11B and 11C). Malar sulcus present. Scape with a well-developed ventral lamina (Figs. 11B and 11D). Upper mesepimeron smooth (Fig. 11F). Hypopygium reaching tip of gaster (Fig. 11A). Relative measurements: Head L: 36, W: 63, H: 55; eye H: 32, L: 25; malar space: 8.5; mouth W: 26; scape L: 16, W: 7; pedicel L: 8, W: 4.5; pedicel plus flagellum L: 52; fu1 L: 6, W: 5.5; fu5 L: 5.5, W: 5.5; clava L: 16, W: 5. Mesosoma L: 63, W: 55, H: 52; mesoscutum L: 25, W: 55; mesoscutellum L: 25, W: 26; propodeum L: 15; fore wing L: 130, W: 63; MV: 29; SV: 15.5; PMV: 30. Metasoma. Petiole L: 20, W: 10; gaster L: 88, W: 50.

**Male**
Unknown.

**Variation**
Body length: 1.65–1.80 mm. Face blue-violet to blue-green. Pronotal collar blue to green. MV 1.8–1.9X SV. Gaster size variable depending on its collapse degree (occasionally strongly compressed laterally).

**Etymology**
The name of the species refers to the peculiar shape of the scape (adjective).

**Distribution**
Cameroon, Gabon, Kenya.

**Biology**
Unknown.

***Scrobesia* Mitroiu & Rasplus, gen. nov.**
urn:lsid:zoobank.org:act:19A729EC-0B71-44A4-98BE-4892719BF808
   (Figs 12, 13)

**Type species**
*Scrobesia acutigaster* Mitroiu & Rasplus, sp. nov., here designated.

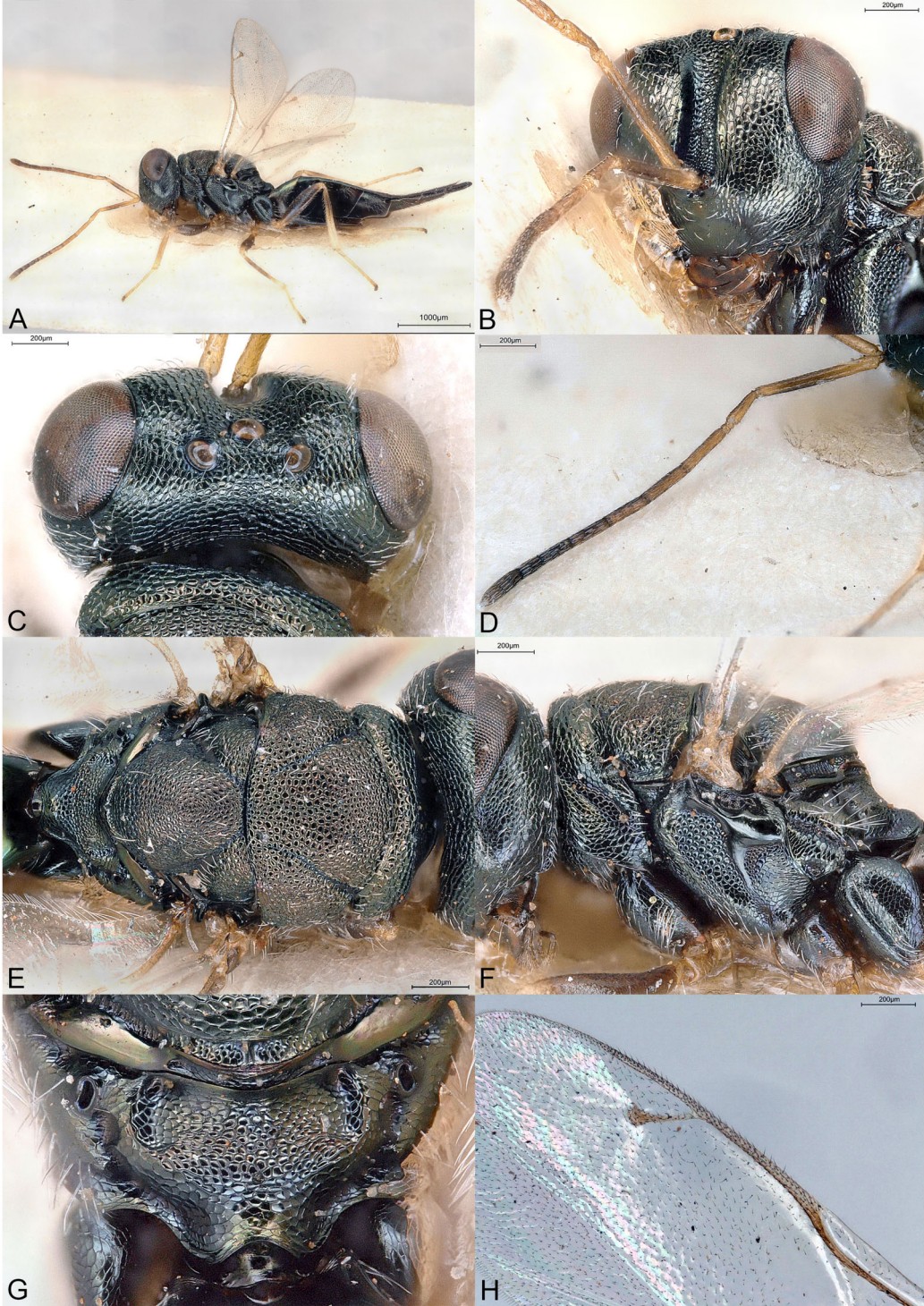

**Figure 12 *Scrobesia acutigaster*.** (A) Female holotype, habitus, lateral. (B) Female holotype, head, frontal. (C) Female holotype, head, dorsal. (D) Female holotype, antenna, lateral. (E) Female holotype, mesosoma, dorsal. (F) Female holotype, mesosoma, lateral. (G) Female holotype, propodeum, dorsal. (H) Female holotype, fore wing, dorsal.

### Diagnosis

**Female**
Antennae inserted below lower ocular line (Figs. 12B and 13B); face protruding at toruli level (Fig. 12A); scrobes very long and deep (Fig. 12B); antenna 11264, long filiform, clava with long narrow strip of micropilosity (Figs. 12D and 13D); notauli almost complete (Figs. 12E and 13E); propodeum with reticulate nucha (Figs. 12G and 13G); gaster sessile, long lanceolate (Figs. 12A and 13A).

### Description

**Female**
Body moderately robust, metallic reflections distinct but dark (Figs. 12A–12C, 12E–12G, 13A–13C, 13E–13G). Body setation short, not very dense (Fig. 13C).

Head trapezoidal in frontal view (Fig. 13B). Clypeal margin broadly emarginate, medially smooth or very finely striate (Fig. 13C). Tentorial pits absent (Fig. 13C). Scrobal depression very deep, without any interantennal crest, stretching to median ocellus (Fig. 12B). Gena not hollowed at mouth corner, posterior edge carinate (Fig. 12B). Malar sulcus very shallow. Eyes moderately large, bare, slightly diverging in lower part (Fig. 13B). Occiput without carina. Head except clypeus reticulate, reticulation coarser in scrobal depression and on vertex (Figs. 12B, 12C, 13B and 13C). Face protruding at antennal insertion (Fig. 12A), toruli far below LOL (Figs. 12B and 13B). Antennal formula 11264 (Figs. 12D and 13D). Both anelli transverse. All funicular segments much longer than broad, narrower than pedicel, sensilla hardly visible. Antennal clava mostly symmetric, but with long narrow strip of microsetation, apex rounded. Mandibles fairly large but not falcate, formula 3:3.

Mesosoma dorsally convex, mostly uniformly reticulate (Figs. 12E, 12F, 13E, 13F). Pronotum short, slightly narrower than mesoscutum (Figs. 12E and 13E). Pronotal collar present, anterior margin abrupt but not carinate (Figs. 12F and 13F). Notauli almost reaching posterior margin of mesoscutum, deep only in anterior third of mesoscutum (Figs. 12E and 13E). Axillae very slightly advanced (Figs. 12E and 13E). Mesoscutellum convex, frenal line absent, frenum defined by a slight to conspicuous colour change (Figs. 12E and 13E). Metascutellum with transverse carina. Propodeum (Figs. 12G and 13G) with basal foveae delimiting short plicae and large convex reticulate nucha. Median carina and costula absent. Median area uniformly reticulate. Propodeal hind corners round. Propodeal spiracles large oval, not touching metanotum, with large postspiracular foveae. Prepectus large, uniformly reticulate, posterior edge raised (Figs. 12F and 13F). Mesopleuron mostly uniformly reticulate except smooth upper mesepimeron (Figs. 12F and 13F). Metapleuron uniformly reticulate, with small ventral depression (Figs. 12F and 13F). Legs slender (Figs. 12A and 13A). Hind coxa dorsally bare except several long setae. Hind tibia with one spur. Fore wing (Figs. 12H and 13H) extensively setose, with basal cell at least partly setose and moderate to small speculum. Marginal vein slender. Stigmal vein much shorter than both marginal and postmarginal veins, the latter shorter than marginal vein. Stigma moderately capitate.

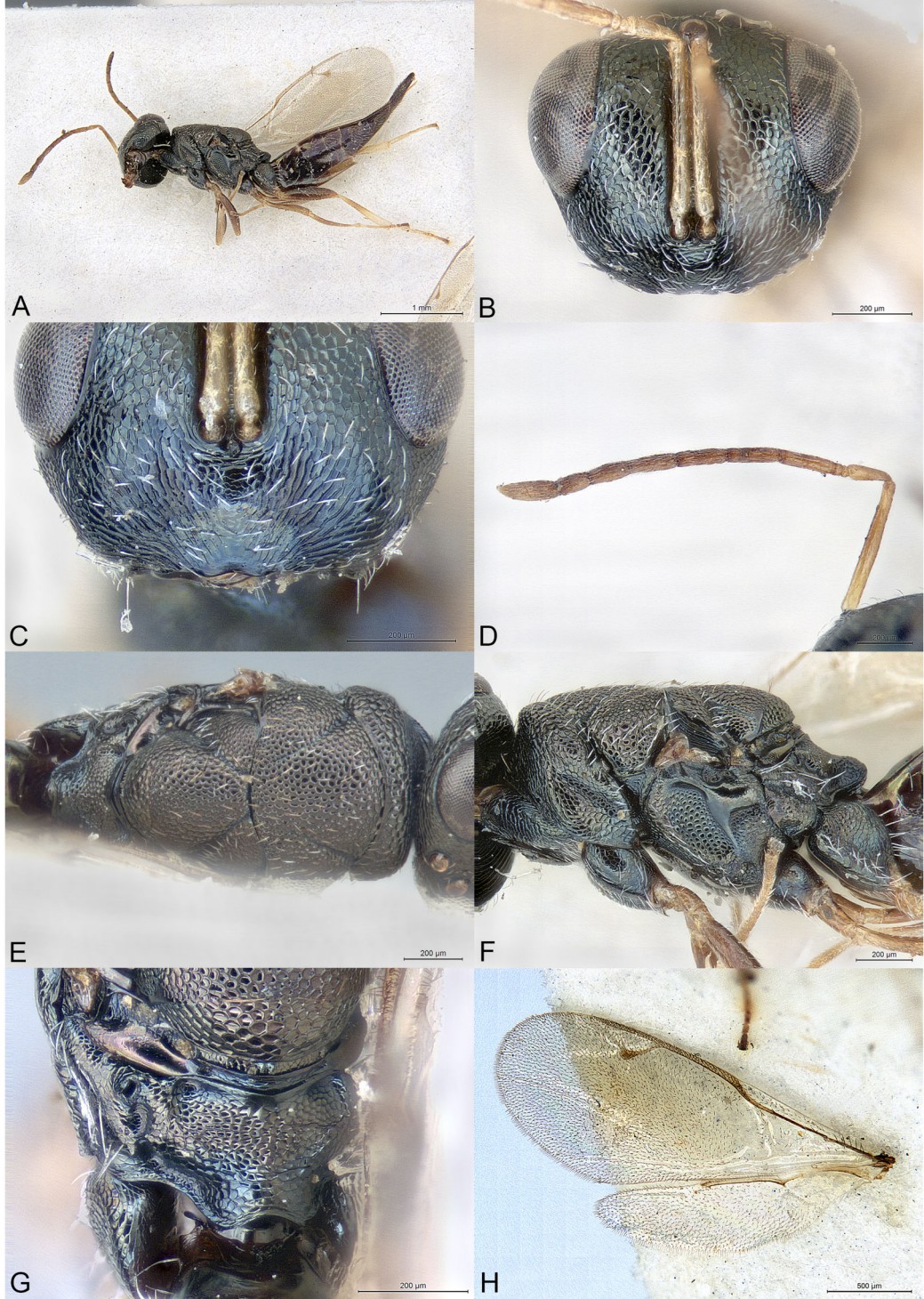

**Figure 13** *Scrobesia pondo.* (A) Female holotype, habitus, lateral. (B) Female holotype, head, frontal. (C) Female paratype, clypeus. (D) Female holotype, antenna, lateral. (E) Female holotype, mesosoma, dorsal. (F) Female holotype, mesosoma, lateral. (G) Female holotype, propodeum, dorsal. (H) Female holotype, fore and hind wings, dorsal.

Metasoma with petiole hardly visible (Figs. 12G and 13G). Gaster long lanceolate, dorsally collapsing (Figs. 12A and 13A). Posterior margin of gt1 medially broadly incised. Hypopygium small, about in the middle of gaster (Figs. 12A and 13A). Cercal setae equal. Ovipositor sheaths slightly exserted (Figs. 12A and 13A).

**Male**
Unknown.

**Etymology**
The generic name (feminine gender) is derived from the face depression (scrobes), which is unusually long and deep in the new genus.

**Relationships**
*Scrobesia* is classified in the subfamily Pteromalinae, tribe Pteromalini based on the following features: (1) antenna with 12 flagellomeres; (2) scapulae not anteriorly exposed by pronotum; (3) notauli incomplete; (4) axillae not strongly advanced; (5) axillulae not enlarged; (6) marginal vein slender; (7) petiole simple (*i.e.*, without anterior flange), very short.

Scrobesia differs from all known genera of Pteromalini in having following combination of characters: (1) antenna filiform, 11264, inserted well below lower ocular line (Figs. 12B and 13B), clava with long narrow strip of micropilosity; (2) face protruding at toruli level (Fig. 12A); (3) scrobes long and deep (Fig. 12B); (4) notauli almost complete (Figs. 12E and 13E); (5) propodeum with large reticulate nucha (Figs. 12G and 13G); (6) gaster sessile, long lanceolate (Figs. 12A and 13A).

In the generic key to Palaearctic Pteromalidae (*Bouček & Rasplus, 1991*), assuming the notauli are complete, *Scrobesia* runs to couplet 66 and *Perniphora* Ruschka, 1923. However, *Scrobesia* greatly differs from the latter genus in the much longer scrobes lacking a high interantennal crest, and many other features, such as the shape of the antenna, clypeus, body sculpture and hind legs. Assuming the notauli are incomplete, *Scrobesia* would run to the first half of couplet 154 due to the "antennal insertion placed on distinct protuberance and very low, lower margins of toruli at least slightly below lower ocular line" (p. 67); however, the "postmarginal vein only slightly longer than the stigmal" is not true for the new genus. Ignoring this last feature would lead to *Tritneptis* Girault, 1908 but the new genus greatly differs from it in many features of the head, antennae, mesosoma and fore wing. The second half of couplet 154 states that "if antennae inserted low then thorax usually strongly flattened dorsally", which is also not true, the mesosoma being clearly convex dorsally. The new genus is also superficially similar to some slender species of *Holcaeus* Thomson, 1878 with long antennae and similar claval structure. *Scrobesia* differs from *Holcaeus* mainly in lacking any ridge or carina on the occiput, much lower antennal insertion and much deeper scrobes. In *Heteroprymna* Graham, 1956 the female antenna is slender filiform, with all funicular segments longer than broad, and the head is slightly protuberant at the level of toruli. *Scrobesia* differs from the latter genus mainly in the much shorter clava, almost complete notauli, different propodeum and a pronotal collar not carinate. *Scrobesia* also shares a few features with *Apelioma* Delucchi, 1956 notably the

long antenna with long narrow strip of micropilosity. However, in the latter genus the antennae are inserted higher, the scrobes are shallow and the propodeum has a costula.

In the generic key to Nearctic Pteromalidae (*Bouček & Heydon, 1997*), assuming the notauli are complete, *Scrobesia* runs to couplet 104 (*Ammeia* Delucchi, 1962 and *Tricyclomischus* Graham, 1956), although the characters "thoracic sculpture weak; body at most 2 mm long" do not apply to the new genus. *Scrobesia* greatly differs from both these genera in numerous features of the head, antennae, mesosoma and wings. If the notauli are considered as incomplete, the new genus would run to couplet 149 and *Arriva Bouček, 1993* although in *Scrobesia* the notauli are not deep. *Arriva* also differs in many other features. Other Holarctic or Nearctic genera with rather deep scrobes are *Xiphydriophagus* Ferrière, 1952 and *Ficicola* Heydon, 1992, respectively. However, they also greatly differ from the new genus in many features of the head, antenna, mesosoma and fore wing.

In the generic key to Australasian Pteromalidae (*Bouček, 1988*), considering the notauli as incomplete (the complete notauli would lead to Miscogastrinae or Pireninae), *Scrobesia* runs to couplet 190 and *Pseudanogmus* Dodd & Girault, 1915. The latter genus differs from *Scrobesia* mainly in having the antenna 11353, shallow scrobes, the clypeus bilobed separated by narrow incision, the propodeum with strong sinuate plicae and median carina, the fore wing infumate, with the postmarginal vein hardly as long as the stigmal vein (*Bouček, 1988*).

In the generic key to Oriental Pteromalidae (*Sureshan & Narendran, 2004*) the new genus runs to couplet 93 and *Mesopolobus* Westwood, 1933 on the account of the low level of toruli. However, the latter genus is very different from *Scrobesia* in the structure of the head, antennae, and propodeum. By continuing to ignore the position of toruli one would get to *Pteromalus*, which also greatly differs from *Scrobesia* in many characters.

Finally, *Scrobesia* has no strong similarities with either of the nine Neotropical genera of Pteromalinae, or the few East Palaearctic or Oriental genera that are not included in any of the above-mentioned keys.

**Key to *Scrobesia* species (females)**
One fore wing hyaline (Fig. 12A), length about 2.8X width, basal cell bare except distal third; MV 2.35X SV and 1.3X PMV; syntergum length 2.25X width … **S. acutigaster Mitroiu & Rasplus, sp. nov.**

- Fore wing broadly and moderately infumate (Fig. 13A), length about 2.5X width, basal cell almost completely setose (Fig. 13H); MV 2.5X SV and 1.5X PMV; syntergum length 1.5X width … **S. pondo Mitroiu, sp. nov.**

**Scrobesia acutigaster Mitroiu & Rasplus, sp. nov.**
urn:lsid:zoobank.org:act:62779131-3F76-4C7B-9DD8-8CFF04A3831B
(Fig. 12)

**Material examined**

**Holotype**

ZIMBABWE: ♀, "Rhodesia, Salisbury, A. Watsham", "80", NHMUK014444292 (NHMUK).

### Description

### Female holotype
Body length: 4.25 mm. Colour as in Fig. 12. Basal cell setose only in distal third. Relative measurements: Head L: 32, W: 61, H: 50; POL: 11; OOL: 8; eye H: 30, L: 23; eye L dorsally: 23; temple L dorsally: 7; malar space: 20; mouth W: 30; scape L: 23, W: 3.5; pedicel L: 10, W: 3; pedicel plus flagellum L: 95; fu1 L: 14, W: 3; fu6 L: 8, W: 4; clava L: 15, W: 4.5. Mesosoma L: 80, W: 49, H: 48; mesoscutum L: 30, W: 49; mesoscutellum L: 30, W: 27; propodeum L: 15; fore wing L: 155, W: 56; MV: 33; SV: 14; PMV: 25. Metasoma L: 155, W: 34; gt1 L: 30, W: 30; gt6 L: 28, W: 23; syntergum L: 27, W: 12.

### Etymology
The name of the species refers to the shape of the gaster (adjective).

### Distribution
Zimbabwe.

### Biology
Unknown.

### *Scrobesia pondo* Mitroiu, sp. nov.
urn:lsid:zoobank.org:act:896851A1-F22C-4F3A-8F96-CEE58F4DBB15
(Fig. 13)

### Material examined

### Holotype

**SOUTH AFRICA:** ♀, "S. Africa. R. E. Turner. Brit. Mus. 1923-547", "Port St. John, Pondoland, Oct. 1923", NHMUK014444293 (NHMUK).

### Paratype

**SOUTH AFRICA:** 1♀, "S. Africa. R. E. Turner. Brit. Mus. 1923–398", "Port St. John, Pondoland, July 10–31.1923", NHMUK014444294 (NHMUK).

### Description

### Female holotype
Body length: 3.5 mm. Colour as in Fig. 13. Basal cell almost completely setose (Fig. 13H). Relative measurements: Head L: 28, W: 53, H: 42; eye H: 25, L: 20; malar space: 17; mouth W: 25; scape L: 29, W: 3; pedicel L: 9.5, W: 3; pedicel plus flagellum L: 84; fu1 L: 11, W: 3; fu6 L: 7, W: 4; clava L: 13.5, W: 4.5. Mesosoma L: 70, W: 44, H: 41; mesoscutum L: 25, W: 44; mesoscutellum L: 25, W: 25; propodeum L: 12; fore wing L: 130, W: 51; MV: 30; SV: 12;

PMV: 20. Metasoma L: 123, W: 25; gt1 L: 20, W: 20; gt6 L: 23, W: 20; syntergum L: 20, W: 13.

**Variation**
Body length: 3.0–3.5 mm.

**Etymology**
The name of the species (noun in apposition) refers to the origin of the species, *i.e.*, Pondoland (natural region of South Africa).

**Distribution**
South Africa.

**Biology**
Unknown.

**Spiniclava Mitroiu & Rasplus, gen. nov.**
urn:lsid:zoobank.org:act:6D3B2D75-8442-4613-8241-C9388CFB7C47
   (Figs. 14, 15)

**Type species**
*Spiniclava baaiensis* Mitroiu, sp. nov., here designated.

**Diagnosis**

**Both sexes**
Clypeal margin virtually straight (Fig. 14C); toruli about level with lower eye margin (Figs. 14C and 15B); pronotal collar virtually as wide as mesoscutum (Fig. 14E), rounded off anteriorly into vertical neck (Figs. 14F and 15D); notauli incomplete (Fig. 14E); propodeum uniformly reticulate except two round basal foveae, with large reticulate nucha (Figs. 14G and 15E); petiole long, smooth, without anterior flange, ventrally embraced by very small extensions of gs1 (Fig. 14G); gastral tergites not unusually enlarged (Figs. 14A and 15A).

**Female**
Antenna strongly clavate, 11354; clava with spicula and strongly asymmetric due to large ventral area of microsetation (Figs. 14D and 15C).

**Male**
Antenna almost filiform, 11264; clava acute but without spicula, only slightly asymmetric (Fig. 14B).

**Description**

**Female**
Body robust, black, with faint metallic reflections (Figs. 14A and 15A). Head and dorsal side of mesosoma except most part of propodeum with white setation (Figs. 14C, 14E and 14F, 15B–15D).

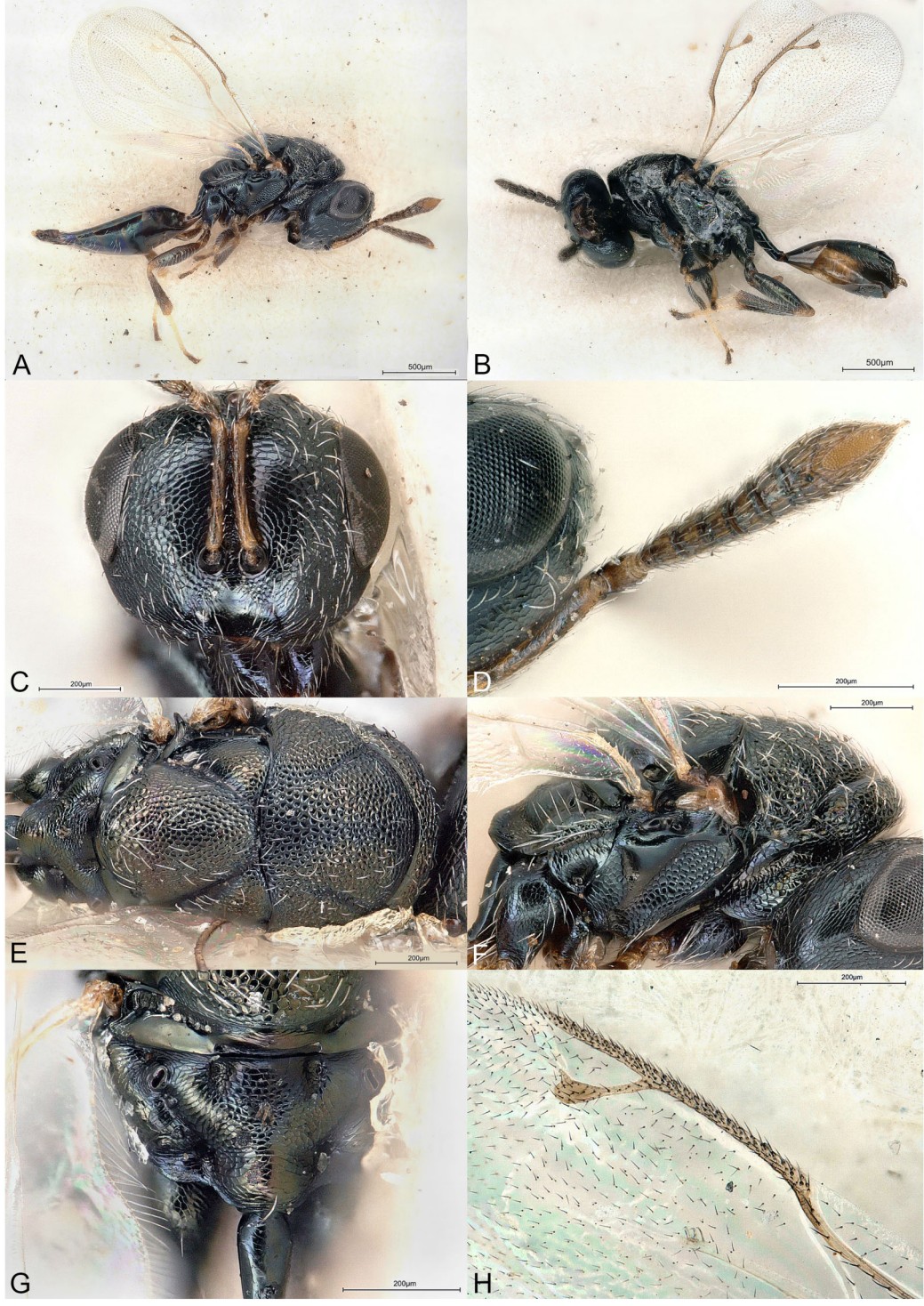

**Figure 14 *Spiniclava baaiensis.*** (A) Female holotype, habitus, lateral. (B) Male allotype, habitus, lateral. (C) Female holotype, head, frontal. (D) Female holotype, antenna, ventral. (E) Male allotype, mesosoma, dorsal. (F) Female holotype, mesosoma, lateral. (G) Male allotype, propodeum, dorsal. (H) Female holotype, fore wing, dorsal.

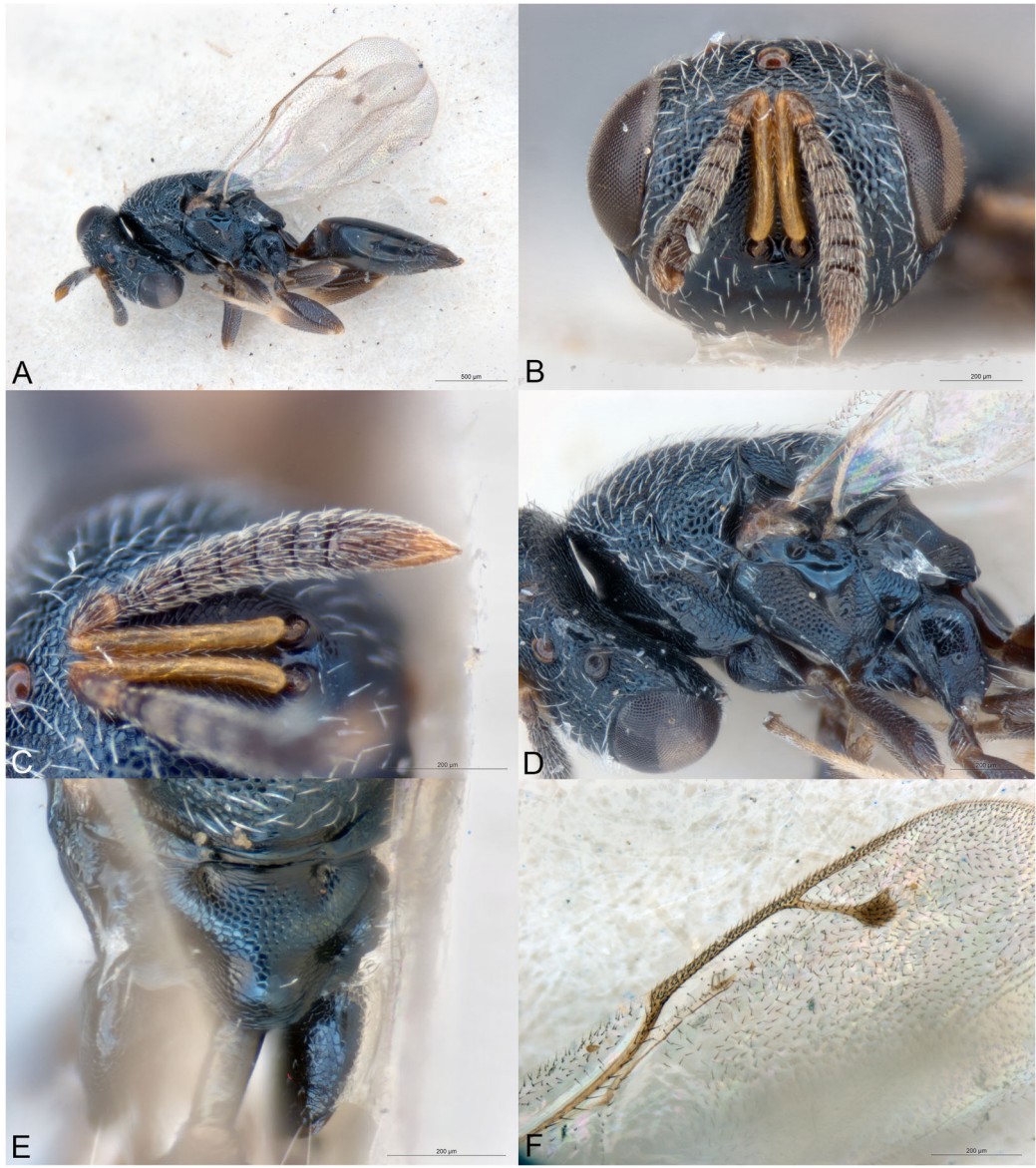

**Figure 15 *Spiniclava setosa*.** (A) Female holotype, habitus, lateral. (B) Female holotype, head, frontal. (C) Female holotype, antenna, lateral. (D) Female holotype, mesosoma, lateral. (E) Female holotype, propodeum, dorsal. (F) Female holotype, fore wing, dorsal.

Head slightly wider than high in frontal view (Figs. 14C and 15B). Clypeal margin virtually straight, without any teeth (Figs. 14C and 15B). Tentorial pits absent. Scrobal depression shallow. Gena not hollowed at mouth corner, weakly carinate near oral fossa. Malar sulcus as a very thin line. Eyes normal (Figs. 14C and 15B). Occiput without carina. Antennal insertion about level with LOL (Figs. 14C and 15B). Antennal formula 11354 (Figs. 14D and 15C). Flagellum with short setae. Anelli transverse, the third larger than any of the previous two. Antennal scape normal. Antennal clava asymmetric, with very large area of microsetation, distal end with spicula. Mandibles not unusually large.

Mesosoma dorsally convex (Figs. 14F and 15D). Pronotum short due to vertical neck (Figs. 14F and 15D). Pronotal collar virtually as wide as mesoscutum, anterior margin rounded (Fig. 14E). Mesoscutum rather short (Fig. 14E). Notauli incomplete, superficial (Fig. 14E). Axillae very slightly advanced (Fig. 14E). Mesoscutellum convex. Frenal line absent (Fig. 14E). Metascutellum short, vertical. Propodeum (Figs. 14G and 15E) with median area funnel-shaped, uniformly reticulate, except two round basal foveae. Propodeal plicae, median carina and costula absent. Nucha large, convex, separated from supracoxal flange by deep pit. Propodeal hind corners round. Propodeal spiracles small oval, clearly separated from posterior margin of metanotum. Prepectus shorter than tegula (Figs. 14F and 15D). Mesopleuron without ventral carina (Fig. 14F). Hind coxa with a few scattered setae on dorsal side. Hind tibia with two spurs, one much longer than the other. Fore wing (Figs. 14H and 15F) hyaline, mostly bare basally. Marginal vein slender. Stigmal vein much shorter than marginal vein. Stigma moderately capitate. Postmarginal vein much shorter than marginal vein and longer than stigmal vein. Marginal fringe present.

Metasoma petiolate, petiole long, smooth, thickest in anterior quarter and gradually becoming thinner posteriorly, basal part ventrally embraced by very small projections of gs1 (Figs. 14F, 14G and 15E). Gaster lanceolate, dorsally flat or convex (Figs. 14A and 15A). Gt1 the longest but not unusually enlarged, its posterior margin entire and posteriorly produced. The following tergites not unusually enlarged, except sometimes the third. Cercal setae equal. Ovipositor sheaths short (Figs. 14A and 15A).

### Male
Differs from the females as follows. Head and mesosoma with slightly stronger and lighter metallic reflections (Fig. 14E). Mandible formula 3:3 (not visible in the female holotype). Antenna filiform (Fig. 14B), formula 11263, the first funicular segment distinctly smaller than the second. Clava symmetric, without visible area of setation on ventral side, apex pointed but without spicula. Flagellum covered by dense and moderately long setae. Gaster much smaller, first tergite occupying about 2/3 gaster length, terminal tergites retracted (Fig. 14B).

### Etymology
The generic name (feminine gender) refers to the spiculate clava of the female.

### Relationships
*Spiniclava* belongs to subfamily Pteromalinae, tribe Pteromalini based on the following features: (1) antenna with 12 flagellomeres (Figs. 14D and 15C); (2) scapulae not anteriorly exposed by pronotum (Fig. 14E); (3) notauli incomplete (Fig. 14E); (4) axillae not strongly advanced (Fig. 14E); (5) axillulae not enlarged (Fig. 14E); (6) marginal vein slender (Figs. 14H and 15F); (7) petiole simple, *i.e.*, without anterior flange, smooth and long (Figs. 14B, 14F, 14G, 15E).

Due to its long tubular petiole, *Spiniclava* is superficially similar to other petiolate Pteromalini. The closest genus seems *Sphegigastrella* Masi, 1917, the shared characters being the presence of five funicular segments, propodeal shape, smooth gastral petiole and shape of gastral tergites. However, *Spiniclava* differs from *Sphegigastrella* in the following

characters: (1) antennae inserted at lower ocular line (Figs. 14C and 15B) (much higher, near center of face in *Sphegigastrella*); (2) antenna distinctly clavate, clava asymmetric, with large area of microsetation and spicula (Figs. 14D and 15C) (in *Sphegigastrella* antenna at most slightly clavate, clava usually symmetric, with at most a small area of microsetation, always without spicula); (3) gena not hollowed at mouth corner (gena at least slightly hollowed in *Sphegigastrella*); (4) central part of clypeal margin not projecting ventrally, straight (Figs. 14C and 15B) (in *Sphegigastrella* central part of clypeal margin slightly projecting ventrally, projection usually slightly emarginate); (5) pronotal collar long, virtually as wide as mesoscutum, pronotal neck vertical (Figs. 14F and 15D) (collar much shorter and narrower in *Sphegigastrella*, collar neck with a distinct slope).

In the generic key to Palaearctic Pteromalidae (*Bouček & Rasplus, 1991*) *Spiniclava* would run to couplet 86 and *Isocyrtus* Walker, 1833 due to its smooth shiny petiole. However, it differs from *Isocyrtus* in the antennal structure (with six funicular segments, clava symmetric, without spicula in *Isocyrtus*), clypeal shape (with two broad teeth, emarginate between them in *Isocyrtus*), pronotal shape (long on sides but much narrower than mesoscutum in *Isocyrtus*), propodeum shape (with subparallel plicae and shorter nucha in *Isocyrtus*), and longer petiole. Ignoring the lack of sculpture on the petiole, *Spiniclava* runs to couplet 93 (*Halticoptera* Spinola, 1811 and *Eurydinota* Förster, 1878). Of these two genera *Spiniclava* is more similar to *Eurydinota* regarding the clypeal margin and propodeum shape; however, it greatly differs from this genus in the shape of the antenna, pronotal collar, petiole and gaster. The large pronotal collar of *Spiniclava* somehow resembles that of *Syntomopus* Walker, 1833, but its anterior corners are clearly round, not rectangular; moreover, *Spiniclava* greatly differs in the shape of mesosoma (not flattened), shape of the clypeal margin (without central tooth), number of funicular segments (less than six), antennal insertion (less high), and petiole structure (not reticulate). In *Syntomopus crassicornis* (Szelényi, 1970) the antennal clava bears a short spine, but other characters are very similar to other species of *Syntomopus* and thus different from *Spiniclava*. Another genus with long petiole and wide pronotal collar is *Paracarotomus*; *Spiniclava* differs from it mainly in the antennal shape and structure, the less high mesosoma, without a distinct shelf between pro- and mesocoxa, the less strong genal carina, and the different shape of the gastral tergites. From both *Sphegigaster* Spinola, 1811 and *Cyrtogaster* Walker, 1833, the new genus differs in many characters, such as the shape of the clypeal margin, antenna, propodeum, petiole, and gastral tergites. For differences between *Spiniclava* and *Callitula* Spinola, 1811, see below.

In the generic key to Nearctic Pteromalidae (*Bouček & Heydon, 1997*) the new genus runs to couplet 134 and *Miristhma Bouček, 1993*. *Spiniclava* differs from this genus mostly in the shape of the antennal clava (without a spicula in *Miristhma*), antennal insertion (well above lower ocular line in *Miristhma*), shape of gena (slightly hollowed near mouth corner in *Miristhma*), pronotal collar (short, more or less carinate in *Miristhma*), and propodeum (horizontal and with a long nucha, which is constricted before apex in *Miristhma*).

In the generic key to Australasian Pteromalidae (*Bouček, 1988*) *Spiniclava* runs to couplet 284 but fails to fit in both halves of the couplet *i.e.*, the petiole is more than twice as

long as broad, but the third gastral tergite is not unusually large and convex. Ignoring the length of the petiole leads to couplet 285 (*Delisleia* and *Aiemea Bouček, 1988*). *Spiniclava* greatly differs from both these genera in many characters such as the shape of antenna, clypeal margin, gena, and petiole length. The general shape of propodeum and petiole are somewhat similar to those of *Merismomorpha* Girault, 1913 (also present in Africa). *Merismomorpha* greatly differs from *Spiniclava* in the shape of the antennal clava (only slightly asymmetric and without spicula), clypeal margin (median part distinctly produced), pronotum (much narrower than mesoscutum), propodeal sculpture (basal foveae with distinct posterior sulci), and the extensions of the first gastral sternite (much larger and laterally embracing the posterior part of petiole). The presence of tree anelli, pointed clava and large reticulate nucha are shared with *Callitula*. The later genus differs from *Spiniclava* mainly in the shape of antennae (filiform), pronotum (distinctly narrower than mesoscutum), petiole (much shorter than propodeum), and extensions of first gastral sternite (larger, laterally embracing the posterior part of petiole).

In the generic key to Oriental Pteromalidae (*Sureshan & Narendran, 2004*) *Spiniclava* runs to couplet 58 and *Merismomorpha*. For the main differences between the two genera, see above.

Among the Neotropical genera not included in any identification key, *Spiniclava* most closely resembles *Notoprymna De Santis, 1988* by the presence of a long petiole, spiculate clava, and large pronotum. The latter genus differs from *Spiniclava* at least in having the following characters: (1) antenna 11263, clava symmetric; (2) notauli complete, well impressed; (3) mesoscutellum with distinct frenal line; (4) propodeal plicae present; (5) first gastral tergite the largest, occupying half the length of gaster (*De Santis, 1988*).

There are three East Palearctic genera of Pteromalinae that exhibit a long petiole: *Amblyharma* Huang & Tong, 1993, *Paroxyharma* Huang & Tong, 1993, and *Sorosina* Dzhanokmen, 1993; according to their original descriptions, they all differ from *Spiniclava* in many characters of the antenna, head, mesosoma and gaster.

**Key to *Spiniclava* species (females)**
One fore wing (Fig. 14H): ventral side of costal cell sparsely setose, ventral side of disc with 2–3 rows of admarginal setae beyond marginal vein, basal cell (except basal and cubital folds) with 0–3 setae, MV 2.1–2.3X SV; prepectus and metapleuron with shallow reticulation (Fig. 14F); gt3 occupying about 1/5-1/6 gaster length; antenna with fu1-2 longer than wide, fu3 quadrate (Fig. 14D); tibiae almost completely dark, extremities brown (Fig. 14A) … **S. baaiensis Mitroiu & Rasplus, sp. nov.**

- Fore wing (Fig. 15F): ventral side of costal cell densely setose, ventral side of disc with 5–6 rows of admarginal setae beyond marginal vein, basal cell (except basal and cubital folds) with 5–9 setae, MV 1.8–2.0X SV; prepectus and metapleuron with strong reticulation (Fig. 15D); gt3 occupying about 1/3–1/4 gaster length; antenna with fu1–2 quadrate, fu3 transverse (Fig. 15C); tibiae less extensively dark, extremities pale yellow (Fig. 15A) … **S. setosa Mitroiu, sp. nov.**

*Spiniclava baaiensis* Mitroiu & Rasplus, sp. nov.
urn:lsid:zoobank.org:act:C269B343-5E34-4754-BD37-E3B30595EBA4
  (Fig. 14)

**Material examined**

**Holotype**

**SOUTH AFRICA:** ♀, "S. Africa. R. E. Turner. Brit. Mus. 1921–294", "Mossel Bay, Cape Province. June 1921.", NHMUK014444295 (NHMUK).

**Allotype**

**SOUTH AFRICA:** ♂, "S. Africa. R. E. Turner. Brit. Mus. 1922–25", "Mossel Bay, Cape Province. Dec. 1921.", NHMUK014444296 (NHMUK).

**Additional paratypes**

**SOUTH AFRICA:** 1♀, "S. Africa. R. E. Turner. Brit. Mus. 1921–294", "Mossel Bay, Cape Province. 1–3. vii. 1921.", NHMUK014444297 (NHMUK).

**Description**

**Female holotype**
Body length: 2.25 mm. Colour as in Figs. 14A, 14C, 14D, 14F and 14H. Antennal scape not reaching lower edge of median ocellus (Fig. 14C). Antenna (Fig. 14D) with fu1–2 longer than wide, fu3 quadrate, fu4–5 transverse. Microsetation area on ventral side of clava occupying about 2/3 claval length. Length of pedicel plus flagellum shorter than head width. Clypeal area almost smooth, shiny, fine striation visible only on sides (Fig. 14C). Rest of the head and dorsal side of mesosoma reticulate (Figs. 14C and 14E). Prepectus and metapleuron shallowly reticulate (Fig. 14F). Mesopleuron reticulate except large smooth triangular area under wings bases (Fig. 14E). Fore wing with basal cell having 1–3 setae, basal and cubital folds setose (most setae broken). Fore wing disc moderately setose, speculum reaching parastigma (Fig. 14H). Ventral side of fore wing with 2–3 rows of admarginal setae beyond marginal vein (Fig. 14H). Ventral side of costal cell sparsely setose, with one row of setae near anterior margin and some additional setae in distal part. Gaster narrower and only slightly longer than mesosoma. Relative measurements: Head L: 25, W: 55, H: 45; eye H: 26, L: 19; malar space: 16; mouth W: 27; scape L: 24, W: 3; pedicel L: 5, W: 3; pedicel plus flagellum L: 42; fu1 L: 4.5, W: 4; fu5 L: 4, W: 6; clava L (including spicula): 15, W: 6.5. Mesosoma L: 74, W: 46, H: 43; mesoscutum L: 29, W: 46; mesoscutellum L: 29, W: 26; propodeum L: 17; fore wing L: 120, W: 56; MV: 25; SV: 11; PMV: 14. Metasoma. Petiole L: 17, W: 6.5; gaster L: 79, W: 32; gt1 L: 27, W: 32; gt3 L: 12, W: 33; gt6 L: 10, W: 23; syntergum L: 8, W: 12.

**Male allotype**
Differs from the female as follows. Colour as in Figs. 14B, 14E and 14G. Flagellum filiform, with longer setae (Fig. 14B). Fu1 slightly transverse, shorter and narrower than fu2. Fu2–4

quadrate, fu5-6 slightly transverse. Fore wing more sparsely setose: basal cell with fewer setae and speculum larger, reaching proximal end of marginal vein and extending as a narrow bare strip to stigmal vein and thus admarginal setae more visible. Petiole slightly longer than propodeum (Fig. 14B). Gaster much shorter and narrower than mesosoma (Fig. 14B), length about 2.1X width. Gt1 triangular, much longer than wide, following tergites partly to completely retracted.

### Variation

### Female
Body length: 2.25–2.50 mm. Basal cell (except basal and cubital folds) with 0–3 setae. MV 2.1–2.3X SV. Metasoma with gt3 occupying about 1/5–1/6 gaster length.

### Etymology
The name of the species (adjective) is a reference to its type locality, Mossel Bay (Afrikaans: Mosselbaai).

### Distribution
South Africa.

### Biology
Unknown.

### *Spiniclava setosa* Mitroiu, sp. nov.
urn:lsid:zoobank.org:act:13A9CD8C-AAA0-42FA-9456-E8704C0BCCCD
(Fig. 15)

### Material examined

### Holotype

**SOUTH AFRICA:** ♀, "S. Africa. R. E. Turner. Brit. Mus. 1923-369", "Port St. John, Pondoland. July 1–9. 1923", NHMUK014444298 (NHMUK).

### Paratypes

**SOUTH AFRICA:** 1♀, "S. Africa. R. E. Turner. Brit. Mus. 1923-332", "Port St. John, Pondoland. May 15–31. 1923", NHMUK014444299 (NHMUK).

### Description

### Female holotype
Body length: 2.5 mm. Colour as in Fig. 15. Antennal scape not reaching lower edge of median ocellus (Fig. 15B). Antenna (Fig. 15C) with fu1–2 quadrate, fu3–5 transverse. Microsetation area on ventral side of clava occupying more than 2/3 claval length (difficult to asess when clava is collapsed). Length of pedicel plus flagellum shorter than head width. Clypeal area almost smooth, shiny, fine striation visible only on sides. Rest of the head and dorsal side of mesosoma reticulate (Fig. 15B). Prepectus and metapleuron reticulate

(Fig. 15D). Mesopleuron reticulate except large smooth triangular area under wings bases (Fig. 15D). Fore wing with basal cell having 8–9 setae, basal and cubital folds setose. Fore wing disc densely setose, speculum reaching parastigma (Fig. 15F). Ventral side of fore wing with 5–6 rows of admarginal setae beyond marginal vein (Fig. 15F). Ventral side of costal cell densely setose, with several rows of setae. Gaster narrower and only slightly longer than mesosoma. Relative measurements: Head L: 27, W: 66, H: 53; eye H: 32, L: 22; malar space: 20; mouth W: 31; scape L: 27, W: 4; pedicel L: 6, W: 4; pedicel plus flagellum L: 50; fu1 L: 5, W: 5; fu5 L: 4.5, W: 8; clava L (including spicula): 16, W: 8. Mesosoma L: 84, W: 53, H: 48; mesoscutum L: 35, W: 53; mesoscutellum L: 29, W: 30; propodeum L: 20; fore wing L: 130, W: 65; MV: 29; SV: 16; PMV: 21. Metasoma. Petiole L: 16, W: 6; gaster L: 88, W: 37; gt1 L: 25, W: 25; gt3 L: 27, W: 37; gt6 L: 9, W: 25; syntergum L: 10, W: 15.

### Etymology
The name of the species refers to the densely setose wings of the species (adjective).

### Variation

### Female
Body length: 2.50–2.75 mm. Basal cell (except basal and cubital folds) with 5–9 setae. MV 1.8–2.0X SV. Metasoma with gt3 occupying about 1/3–1/4 gaster length.

### Distribution
South Africa.

### Biology
Unknown.

## CONCLUSIONS

This study adds seven new genera of Chalcidoidea to the Afrotropical fauna, all described herein: one in the family Cerocephalidae, one in Epichrysomallidae, one in Pirenidae, and four in Pteromalidae. In total, 13 new species are described, one for each Cerocephalidae, Epichrysomallidae and Pirenidae, and 10 in Pteromalidae. The material examined originates from nine African countries, including Madagascar.

Except for two species, their biology is unknown. *Delvareus dicranostylae* (Epichrysomallidae) is probably a gallmaker on *Ficus dicranostyla*, while *Pilosalis barbatulus* (Pteromalidae) was reared from an unknown host, whose remains suggest mealybugs (Hemiptera: Pseudococcidae) or similar hemipterans.

## ACKNOWLEDGEMENTS

We thank all curators of the institutions where the material is deposited for loans of specimens and assistance during research visits. We are indebted to John Noyes, Petr Janšta and an anonymous reviewer for their valuable comments and suggestions.

### Funding

Financial support was provided by the Romanian Ministry of Research, Innovation and Digitization, within Program 1–Development of the national RD system, Subprogram 1.2–Institutional Performance, RDI excellence funding projects, contract no. 11PFE/30.12.2021, and the European Commission through the Synthesys grant BE-TAF-464 and the Synthesys+ grant GB-TAF-TA4-016, to Mircea-Dan Mitroiu. This work is based upon support received from the South African National Research Foundation (grants GUN 61497 and GUN 98115) to Simon van Noort. The funders had no role in study design, data collection and analysis, decision to publish, or preparation of the manuscript.

### Grant Disclosures

The following grant information was disclosed by the authors:
Romanian Ministry of Research, Innovation and Digitization: 11PFE/30.12.2021.
European Commission: BE-TAF-464 and GB-TAF-TA4-016.
South African National Research Foundation: GUN 61497 and GUN 98115.

### Competing Interests

The authors declare that they have no competing interests.

### Author Contributions

- Mircea-Dan Mitroiu analyzed the data, prepared figures and/or tables, authored or reviewed drafts of the article, and approved the final draft.
- Jean-Yves Rasplus analyzed the data, prepared figures and/or tables, authored or reviewed drafts of the article, and approved the final draft.
- Simon van Noort analyzed the data, authored or reviewed drafts of the article, and approved the final draft.

### Data Availability

Material examined

Milokoa villemantae Mitroiu, sp. nov.
urn:lsid:zoobank.org:act:B78D19D2-C6FE-49F5-811F-FC6126E964D6
EY36195 (MNHN); MICO-2023-1 (MICO).

Delvareus dicranostylae Rasplus, Mitroiu & van Noort sp. nov.
urn:lsid:zoobank.org:act:5E6F6D56-470C-4FA5-B3FA-C95DB47D8664
JRAS01442_0101; JRAS01442_0102; JRAS01442_0103; JRAS01442_0104; JRAS01442_0105; JRAS01442_0106; JRAS01442_0107; JRAS01442_0108; JRAS01442_0109; JRAS01442_0110; JRAS01442_0111 (CBGP).

Afrothopus georgei Mitroiu, sp. nov.
urn:lsid:zoobank.org:act:CC8CB6B6-F273-4BC9-B0D5-0932DDA1114F
NHMUK014444237; NHMUK014444238; NHMUK014444239 (NHMUK); MICO-2023-2 (MICO).

Kerangania nuda Mitroiu, sp. nov.

urn:lsid:zoobank.org:act:5ED1BA05-4E04-4C51-AE48-71F44423B309

NHMUK014444241; NHMUK014444242 (NHMUK).

Pilosalis barbatulus Mitroiu, sp. nov.

urn:lsid:zoobank.org:act:82A2EAA5-A2D1-4C1A-9DB4-1331F3DCCBA7

NHMUK014444243; NHMUK014444244; NHMUK014444246; NHMUK014444247; NHMUK014444248; NHMUK014444249; NHMUK014444250; NHMUK014444251; NHMUK014444252; NHMUK014444253; NHMUK014444254; NHMUK014444255; NHMUK014444256; NHMUK014444257; NHMUK014444258; NHMUK014444259; NHMUK014444260; NHMUK014444261; NHMUK014444262; NHMUK014444263; NHMUK014444264; NHMUK014444265; NHMUK014444266; NHMUK014444267; NHMUK014444269; NHMUK014444270; NHMUK014444271; NHMUK014444272; NHMUK014444273; NHMUK014444274; NHMUK014444275; NHMUK014444276; NHMUK014444277; NHMUK014444278; NHMUK014444279; NHMUK014444280; NHMUK014444281; NHMUK014444282; NHMUK014444283; NHMUK014444284 (NHMUK); MICO-2023-3; MICO-2023-4 (MICO).

Pilosalis bouceki Mitroiu & Rasplus, sp. nov.

urn:lsid:zoobank.org:act:0902BB2C-AE87-4780-99EB-2EED1BAF1BB5

NHMUK014444285; NHMUK014444286; NHMUK014444287; NHMUK014444288; NHMUK014444289 (NHMUK); JRAS08824_0101 (CBGP); 27493; 27494 (NMPC).

Pilosalis eurys Mitroiu & van Noort, sp. nov.

urn:lsid:zoobank.org:act:D3A6FA39-2003-40D5-8E19-27639D6AAD33

27495; 27496 (NMPC); SAM-HYM-P078965; SAM-HYM-P082130; SAM-HYM-P078970; CAR01-Y28, CAR01-Y34, CAR01-Y38, CAR01-Y40, CAR01-Y43, CAR01-Y50, SAM-HYM-P078966; SAM-HYM-P078967; SAM-HYM-P078968; SAM-HYM-P078969, SAM-HYM-P082126, SAM-HYM-P082127, SAM-HYM-P082131, SAM-HYM-P082580, SAM-HYM-P082581; SAM-HYM-P082583, SAM-HYM-P082584 (SAMC).

Pilosalis minutus Mitroiu, sp. nov.

urn:lsid:zoobank.org:act:6259AE7E-AED9-4574-97A0-2A77B7E99565

NHMUK014444290 (NHMUK).

Pilosalis platyscapus Mitroiu, Rasplus & van Noort, sp. nov.

urn:lsid:zoobank.org:act:1B2260C7-A006-4800-B62A-4D8F0270B6A2

NHMUK014444291 (NHMUK); SAM-HYM-P0023796 (SAMC); JRAS08825_0101 (CBGP).

Scrobesia acutigaster Mitroiu & Rasplus, sp. nov.

urn:lsid:zoobank.org:act:62779131-3F76-4C7B-9DD8-8CFF04A3831B

NHMUK014444292 (NHMUK).

Scrobesia pondo Mitroiu, sp. nov.

urn:lsid:zoobank.org:act:896851A1-F22C-4F3A-8F96-CEE58F4DBB15

NHMUK014444293; NHMUK014444294 (NHMUK).

Spiniclava baaiensis Mitroiu & Rasplus, sp. nov.

urn:lsid:zoobank.org:act:C269B343-5E34-4754-BD37-E3B30595EBA4

NHMUK014444295; NHMUK014444296; NHMUK014444297 (NHMUK).

Spiniclava setosa Mitroiu, sp. nov.
urn:lsid:zoobank.org:act:13A9CD8C-AAA0-42FA-9456-E8704C0BCCCD
NHMUK014444298; NHMUK014444299 (NHMUK).

## New Species Registration

The following information was supplied regarding the registration of a newly described species:

Publication LSID: urn:lsid:zoobank.org:pub:8A49E9CD-1FD9-4B3A-8285-CAA71CEE7A46

Afrothopus: urn:lsid:zoobank.org:act:6DCDAD4A-03E5-4B98-814B-355B8A93B97E

Delvareus: urn:lsid:zoobank.org:act:D4085947-BE16-44F9-9502-692B31FDA24F

Kerangania: urn:lsid:zoobank.org:act:DC6BE0D6-F230-4E97-A057-F814DC691140

Milokoa: urn:lsid:zoobank.org:act:8079D822-B567-4C69-844D-3329C6654618

Pilosalis: urn:lsid:zoobank.org:act:F2772385-9073-45D7-BA3C-3B2AA0791E62

Scrobesia: urn:lsid:zoobank.org:act:19A729EC-0B71-44A4-98BE-4892719BF808

Spiniclava: urn:lsid:zoobank.org:act:6D3B2D75-8442-4613-8241-C9388CFB7C47

Afrothopus georgei: urn:lsid:zoobank.org:act:CC8CB6B6-F273-4BC9-B0D5-0932DDA1114F

Delvareus dicranostylae: urn:lsid:zoobank.org:act:5E6F6D56-470C-4FA5-B3FA-C95DB47D8664

Kerangania nuda: urn:lsid:zoobank.org:act:5ED1BA05-4E04-4C51-AE48-71F44423B309

Milokoa villemantae: urn:lsid:zoobank.org:act:B78D19D2-C6FE-49F5-811F-FC6126E964D6

Pilosalis barbatulus: urn:lsid:zoobank.org:act:82A2EAA5-A2D1-4C1A-9DB4-1331F3DCCBA7

Pilosalis bouceki: urn:lsid:zoobank.org:act:0902BB2C-AE87-4780-99EB-2EED1BAF1BB5

Pilosalis eurys: urn:lsid:zoobank.org:act:D3A6FA39-2003-40D5-8E19-27639D6AAD33

Pilosalis minutus: urn:lsid:zoobank.org:act:6259AE7E-AED9-4574-97A0-2A77B7E99565

Pilosalis platyscapus: urn:lsid:zoobank.org:act:1B2260C7-A006-4800-B62A-4D8F0270B6A2

Scrobesia acutigaster: urn:lsid:zoobank.org:act:62779131-3F76-4C7B-9DD8-8CFF04A3831B

Scrobesia pondo: urn:lsid:zoobank.org:act:896851A1-F22C-4F3A-8F96-CEE58F4DBB15

Spiniclava baaiensis: urn:lsid:zoobank.org:act:C269B343-5E34-4754-BD37-E3B30595EBA4

Spiniclava setosa: urn:lsid:zoobank.org:act:13A9CD8C-AAA0-42FA-9456-E8704C0BCCCD

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
