# Peer review of "New genera of Afrotropical Chalcidoidea (Hymenoptera: Cerocephalidae, Epichrysomallidae, Pirenidae and Pteromalidae)"

_PeerJ, doi:10.7717/peerj.16798_

## Round 0.1 · original submission · Minor Revisions

Please, provide minor revisions requested by the reviewers prior to the final acceptance.

·

Basic reporting

This is a straight-forward description of seven new genera and 13 new species of Pteromalidae, the names of which are required for a planned treatment of the genera of Afrotropical of this family. An identification key to the genera of this family is badly needed to enable students of the group to attempt their identification in this region. To date there is no key available for this purpose and therefore I think thus contribution is very suitable for publication.

I can find very little to fault in the MS. It is clearly and concisely written with a detailed description of the methods used. To my mind the literature cited is sufficient, although one citation is missing from the list of references at the end (I have marked this in the text).

Experimental design

The MS is original and the work has been done to a high standard. Diagnostic characters are provided in all cases. The illustrations are numerous and in most cases are excellent. Those that are not excellent are made from older, museum specimens and often are quite dirty and thus acceptable.

Validity of the findings

The authors proved detailed comparsions with similar or related genera that occur in any part of the world.

Additional comments

I have made some comments on the MS which the authors should consider. Some are best defined as “nit-picky” (e.g. the inconsistent use of singular and plural for some body parts), but I think would improve the MS if they could be incorporated if at all possible. There is also some questions with regards to the depositories of some of the specimens (marked on the MS) and there are also one or two spelling errors and inconsistencies in the English used (marked on the MS).

Reviewer 2 ·

Basic reporting

This research provides a very good addition to the African fauna of this group of insects that needs many studies in that region.
Therefore, the additions provided are very good.
The identfication of the genera and species was comprehensive and well, and an excellent comparison was made with all related taxa, and therefore the taxonomic definitions are correct.

Experimental design

Perfect
It relied on a good number of specimens, and they were well examined

The photos are very clear and well taken, clearly showing the important taxonomic characters

Validity of the findings

Very good

Additional comments

Thank you very much

·

Basic reporting

no comment

Experimental design

no comment

Validity of the findings

no comment

Additional comments

The manuscript entitled "New genera of Afrotropical Chalcidoidea (Hymenoptera: Cerocephalidae, Epichrysomallidae, Pirenidae and Pteromalidae) " is very well written with minimum inaccuracies and coupled with many very useful and high-quality images of all taxa. When more than one species are described, authors add also provide key to the species. Every new genus is properly discussed in terms of potential morphological relationships with other, similar, in the past described, taxa from different parts of the world.

After a few minor changes suggested by me in the attached pdf version of the manuscript, the manuscript will be ready to accept for publication in PeerJ.

With best regards,
Petr Janšta

---

## Round 0.2 · accepted · Accept

The paper can be accepted at this stage.